# A Deep Generative Learning Approach for Two-stage Adaptive Robust Optimization

**Aron Brenner, Rahman Khorramfar, Jennifer Sun, & Saurabh Amin**
Laboratory for Information & Decision Systems
Massachusetts Institute of Technology
Cambridge, MA 02139, USA
`{abrenner,khorram,jennisun,amins}@mit.edu`

## Abstract

Two-stage adaptive robust optimization (ARO) is a powerful approach for planning under uncertainty, balancing first-stage decisions with recourse decisions made after uncertainty is realized. To account for uncertainty, modelers typically define a simple uncertainty set over which potential outcomes are considered. However, classical methods for defining these sets unintentionally capture a wide range of unrealistic outcomes, resulting in ineffective and costly planning in anticipation of unlikely contingencies. In this work, we introduce AGRO, a solution algorithm that performs adversarial generation for two-stage adaptive robust optimization using a variational autoencoder. AGRO generates high-dimensional contingencies that are simultaneously adversarial and realistic, improving the robustness of first-stage decisions at a lower planning cost than standard methods. To ensure generated contingencies lie in high-density regions of the uncertainty distribution, AGRO defines a tight uncertainty set as the image of "latent" uncertainty sets under the VAE decoding transformation. Projected gradient ascent is then used to maximize recourse costs over the latent uncertainty sets by leveraging differentiable optimization methods. We demonstrate the cost-efficiency of AGRO by applying it to both a synthetic production-distribution problem and a real-world power system expansion setting. We show that AGRO outperforms the standard column-and-constraint algorithm by up to $1.8\%$ in production-distribution planning and up to $8\%$ in power system expansion.

## 1 Introduction

Growing interest in data-driven stochastic optimization has been facilitated by the increasing availability of more granular data for a wide range of settings where decision-makers hedge against uncertainty and shape operational risks through effective planning. Adaptive robust optimization (ARO) – also known as adjustable or multi-stage robust optimization – is a class of stochastic optimization models with applications spanning optimization of industrial processes (Gong & You, 2017), transportation systems (Xie et al., 2020), and energy systems planning (Bertsimas et al., 2012). In this work, we aim to solve data-driven two-stage ARO problems under high-dimensional uncertainty. These problems take the form:

$$\min_{\boldsymbol{x} \in \mathcal{X}} \left\{ \boldsymbol{c}^{\top} \boldsymbol{x} + \max_{\boldsymbol{\xi} \in \mathcal{U}} \min_{\boldsymbol{y} \in \mathcal{Y}(\boldsymbol{\xi}, \boldsymbol{x})} \boldsymbol{d}(\boldsymbol{\xi})^{\top} \boldsymbol{y} \right\}, \tag{1}$$

where $\boldsymbol{x}$ denotes "here-and-now" first-stage decisions and $\boldsymbol{y}$ denotes the "wait-and-see" recourse decisions made after the random variable $\boldsymbol{\xi} \in \mathbb{R}^D$ is realized. The set $\mathcal{U} \subset \mathbb{R}^D$ represents the *uncertainty set*, encompassing all possible realizations of $\boldsymbol{\xi}$ that must be accounted for when identifying the optimal first-stage decisions $\boldsymbol{x}$. Here, we assume that the first-stage decisions $\boldsymbol{x}$ are mixed-integer variables taking values in $\mathcal{X}$ while the recourse decisions $\boldsymbol{y}$ take values in the polyhedral set $\mathcal{Y}(\boldsymbol{\xi}, \boldsymbol{x}) = \{\boldsymbol{y} \mid \boldsymbol{B}(\boldsymbol{\xi})\boldsymbol{y} \geq \boldsymbol{b}(\boldsymbol{\xi}) - \boldsymbol{A}(\boldsymbol{\xi})\boldsymbol{x}, \ \boldsymbol{y} \geq \boldsymbol{0}\}$, where $\boldsymbol{d}(\cdot)$, $\boldsymbol{B}(\cdot)$, $\boldsymbol{b}(\cdot)$, and $\boldsymbol{A}(\cdot)$ are affine functions of $\boldsymbol{\xi}$. Additionally, we assume feasibility and boundedness of the recourse problem (i.e. *complete recourse*) for any first-stage decision $\boldsymbol{x} \in \mathcal{X}$ and uncertainty realization $\boldsymbol{\xi} \in \mathcal{U}$.

**Example (Power System Expansion Planning).** As an illustrative application of ARO, consider the problem of capacity expansion planning for a regional generation and transmission system. In this case, $x$ denotes long-term decisions (i.e., installing renewable power plants and transmission lines), which require an upfront investment $c^\top x$. As time unfolds and demand for electricity $\xi$ becomes known, electrical power $y \in \mathcal{Y}(\xi, x)$ can be generated and dispatched to meet demand at a recourse cost of $d(\xi)^\top y$. To ensure robustness against uncertain demand, first-stage decisions balance investment costs with the worst-case recourse costs incurred over the uncertainty set $\mathcal{U}$.

The uncertainty set $\mathcal{U}$ plays a major role in shaping planning outcomes and should be carefully constructed based on the available data. To optimize risk measures such as worst-case costs or value-at-risk, it is desirable to construct $\mathcal{U}$ to be as small as possible while still satisfying probabilistic guarantees of coverage (Hong et al., 2021). However, standard methods for constructing such uncertainty sets can yield highly conservative solutions to problem 1 – i.e., solutions that *over-commit* or otherwise allocate resources ineffectively in anticipation of extreme contingencies – particularly in the case of high-dimensional and irregularly distributed uncertainties for which high-density regions are not well-approximated by conventional (e.g., polyhedral or ellipsoidal) uncertainty sets. While recent works have proposed using deep learning methods to construct tighter uncertainty sets for single-stage robust optimization (Goerigk & Kurtz, 2023; Chenreddy et al., 2022), such approaches have not yet been extended to the richer but more challenging setting of ARO.

We propose AGRO, a solution algorithm that embeds a variational autoencoder (VAE) within a column-and-constraint generation (CCG) scheme to perform adversarial generation of realistic contingencies for two-stage adaptive robust optimization. Our contributions are listed below:

- **Extension of deep data-driven uncertainty sets to ARO.** We propose a formulation for the adversarial subproblem that extends recent approaches for learning tighter uncertainty sets in robust optimization (RO) (Hong et al., 2021; Goerigk & Kurtz, 2023; Chenreddy et al., 2022) to the case of ARO, a richer class of optimization models for planning that allows for recourse decisions to be made after uncertainty is realized.

- **ML-assisted optimization using exact solutions.** In contrast to approaches that train predictive models for recourse costs (Bertsimas & Kim, 2024; Dumouchelle et al., 2024), AGRO optimizes with respect to *exact* recourse costs. Importantly, this eliminates the need to construct a large dataset of solved problem instances for model training.

- **Fast heuristic solution algorithm.** To solve the adversarial subproblem quickly, we utilize a projected gradient ascent (PGA) method for approximate max-min optimization that differentiates recourse costs with respect to VAE-generated uncertainty realizations. By backpropagating through the decoder, we are able to search for worst-case realizations with respect to a latent variable while limiting our search to high-density regions of the underlying uncertainty distribution.

- **Application to ARO with synthetic and historical data.** We apply our solution algorithm to two problems: (1) a production-distribution problem with synthetic demand data and (2) a long-term power system expansion problem with historical supply/demand data. We show that AGRO is able to reduce total costs by up to 1.8% for the production-distribution problem and up to 8% for the power system expansion problem when compared to solutions obtained by the classical CCG algorithm.

## 2 Background

### 2.1 ARO Preliminaries

**Uncertainty Sets.** A fundamental modeling choice in both single-stage RO and ARO is the definition of the uncertainty set $\mathcal{U}$, which captures the range of uncertainty realizations for which the first stage decision $x$ must be robust. In a data-driven setting, the classical approach for defining $\mathcal{U}$ is to choose a class of uncertainty set – common examples include box, budget, or ellipsoidal uncertainty sets (see Appendix A.1) – and to "fit" the set to the data using sample estimates of the mean, covariance, and/or minimum/maximum values. A parameter $\Gamma > 0$ is usually introduced to determine the size of the uncertainty set, and by extension, the desired degree of robustness.

$\Gamma$ is often chosen to ensure a probabilistic guarantee of level $\alpha$, or in the case of problem 1, to ensure:

$$\max_{\boldsymbol{\xi} \in \mathcal{U}} \min_{\boldsymbol{y} \in \mathcal{Y}(\boldsymbol{\xi}, \boldsymbol{x})} \boldsymbol{d}(\boldsymbol{\xi})^\top \boldsymbol{y} \leq \nu \implies \mathbb{P}_{\boldsymbol{\xi} \sim p_{\boldsymbol{\xi}}} \left( \min_{\boldsymbol{y} \in \mathcal{Y}(\boldsymbol{\xi}, \boldsymbol{x})} \boldsymbol{d}(\boldsymbol{\xi})^\top \boldsymbol{y} \leq \nu \right) \geq \alpha, \tag{2}$$

where $\nu$ can be understood as an upper bound for the $\alpha$-value-at-risk (VaR) of the recourse cost. To obtain such a probabilistic guarantee, one typically identifies an appropriate value for $\Gamma$ by leveraging concentration inequalities in the case of single-stage RO (Bertsimas et al., 2021), or more generally, using robust quantile estimates of data "depth" (e.g., Mahalanobis depth) such that $\mathbb{P}_{\boldsymbol{\xi} \sim p_{\boldsymbol{\xi}}}(\boldsymbol{\xi} \in \mathcal{U}) \geq \alpha$ with high probability (Hong et al., 2021).

In order to reduce over-conservative decision-making, one would prefer $\mathcal{U}$ to provide a *tight* approximation of the probabilistic constraint. In other words, it is desirable to make $\mathcal{U}$ as small as possible while still satisfying equation 2. Classical uncertainty sets, however, tend to yield looser approximations of such constraints as the dimensionality of $\boldsymbol{\xi}$ increases (Lam & Qian, 2019). To this end, recent works have proposed learning frameworks for constructing tighter uncertainty sets in single-stage RO settings, which we discuss in Section 2.2.

**Column-and-Constraint Generation.** Once $\mathcal{U}$ is defined, one can apply a number of methods to either exactly (Zeng & Zhao, 2013; Thiele et al., 2009) or approximately (Kuhn et al., 2011) solve the ARO problem. We focus our discussion on the CCG algorithm (Zeng & Zhao, 2013) due to its prevalence in the ARO literature – including its use in recent learning-assisted methods (Bertsimas & Kim, 2024; Dumouchelle et al., 2024) – and because it provides a nice foundation for introducing AGRO in Section 3. The CCG algorithm is an exact solution method for ARO that iteratively identifies "worst-case" uncertainty realizations $\boldsymbol{\xi}^i$ by maximizing recourse costs for a given first-stage decision $\boldsymbol{x}^*$. These uncertainty realizations are added in each iteration to the finite scenario set $\mathcal{S}$, which is used to instantiate the *main problem*:

$$\min_{\boldsymbol{x}, \boldsymbol{y}, \gamma} \quad \boldsymbol{c}^\top \boldsymbol{x} + \gamma \tag{3a}$$

$$\text{s.t.} \quad \boldsymbol{x} \in \mathcal{X}, \tag{3b}$$

$$\boldsymbol{A}(\boldsymbol{\xi}^i)\boldsymbol{x} + \boldsymbol{B}(\boldsymbol{\xi}^i)\boldsymbol{y}^i \geq \boldsymbol{b}(\boldsymbol{\xi}^i), \qquad i = 1, \ldots, |\mathcal{S}| \tag{3c}$$

$$\boldsymbol{y}^i \geq \boldsymbol{0}, \qquad i = 1, \ldots, |\mathcal{S}| \tag{3d}$$

$$\gamma \geq \boldsymbol{d}(\boldsymbol{\xi})^\top \boldsymbol{y}^i, \qquad i = 1, \ldots, |\mathcal{S}| \tag{3e}$$

where $\gamma$ denotes the worst-case recourse cost obtained over $\mathcal{S}$. In each iteration $i$ of the CCG algorithm, additional variables (i.e., columns) $\boldsymbol{y}^i$ and constraints $\boldsymbol{y}^i \in \mathcal{Y}(\boldsymbol{\xi}^i, \boldsymbol{x})$ corresponding to the most recently identified worst-case realization are added to the main problem.

Solving the main problem yields a set of first-stage decisions, $\boldsymbol{x}^*$, which are fixed as parameters in the *adversarial subproblem*:

$$\max_{\boldsymbol{\xi} \in \mathcal{U}} \min_{\boldsymbol{y} \in \mathcal{Y}(\boldsymbol{\xi}, \boldsymbol{x}^*)} \boldsymbol{d}(\boldsymbol{\xi})^\top \boldsymbol{y}. \tag{4}$$

Solving the adversarial subproblem yields a new worst-case realization $\boldsymbol{\xi}^i$ to be added to $\mathcal{S}$. This max-min problem can be solved by maximizing the dual objective of the recourse problem subject to its optimality (i.e., KKT) conditions (Zeng & Zhao, 2013). Despite its prominence in the literature, the KKT reformulation of problem 4 yields a bilinear optimization problem; specifically, one for which the number of bilinear terms scales linearly with the dimensionality of $\boldsymbol{\xi}$. This need to repeatedly solve a large-scale bilinear program can cause CCG to be computationally demanding in the case of high-dimensional uncertainty. To this end, a number of alternative solution approaches – including some that leverage ML methods – have been proposed that approximate recourse decisions in order to yield more tractable reformulations.

## 2.2 Related Work

**Approximate and ML-assisted ARO.** As an alternative to CCG and other exact methods, linear decision rules (LDRs) are commonly used to approximate recourse decisions as affine functions of uncertainty (Kuhn et al., 2011). To better approximate recourse costs using a richer class of

functions, Rahal et al. (2022) propose a "deep lifting" procedure with neural networks to learn piecewise LDRs. Bertsimas & Kim (2024) train decision trees to predict near-optimal first-stage decisions, worst-case uncertainty, and recourse decisions, which are deployed as part of a solution algorithm for ARO. Similarly, Dumouchelle et al. (2024) train a neural network to approximate optimal recourse costs conditioned on first-stage decisions, which they embed as a mixed-binary linear program (Fischetti & Jo, 2018) within a CCG-like algorithm.

While these methods offer a way to avoid the challenging bilinear subproblem in CCG, they come with certain drawbacks. In particular, these methods optimize with respect to an *approximation* of recourse decision-making that is either simplified and endogenized as first-stage variables (in the case of LDRs) or learned in a supervised manner. As such, they *cannot be expected to reduce planning costs over CCG*. Moreover, in the case of (Dumouchelle et al., 2024; Bertsimas & Kim, 2024), a large dataset of recourse problem instances must be collected and solved, which can be computationally demanding. Additionally, these works are focused on improving problem tractability and do not consider the challenge of loose uncertainty sets in high-dimensional settings.

**Data-driven Uncertainty Sets.** Some works have aimed to address this issue of loose uncertainty representation for single-stage RO by learning tighter uncertainty sets from data. One of the earliest such works, Tulabandhula & Rudin (2014), takes a statistical learning approach to building uncertainty sets from data. Hong et al. (2021) provide a method for calibrating $\Gamma$ using measures of "data depth" and propose clustering and dimensionality reduction methods for constructing uncertainty sets. More similarly to our work, Goerigk & Kurtz (2023) introduce a solution algorithm that first constructs a "deep data-driven" uncertainty set as the image of a Gaussian superlevel set under a neural network-learned transformation and then iteratively identifies worst-case realizations by optimizing through the neural network. Chenreddy et al. (2022) extend their approach to construct uncertainty sets conditioned on the observation of a subset of covariates.

Both Goerigk & Kurtz (2023) and Chenreddy et al. (2022) leverage the main result of Fischetti & Jo (2018) to embed a neural network within an adversarial subproblem formulated as a mixed-binary linear program. However, extending this approach to ARO with high-dimensional uncertainty poses significant challenges. Specifically, the lack of a closed-form expression for the recourse cost leads to an adversarial subproblem that becomes a mixed-binary *bilinear* program (see Appendix A.2). While solution algorithms for such problems exist, their convergence will be slow for instances involving a large number of binary variables, as is the case when embedding neural networks. This underscores the need for fast, alternative approaches capable of generating approximate solutions for such bilinear adversarial problems, thereby enabling efficient worst-case analysis with "deep data-driven uncertainty" to extend beyond single-stage RO to the more general case of ARO.

# 3 Methodology

We now describe AGRO, a CCG-like solution algorithm that performs adversarial generation for robust optimization. As illustrated in Figure 1, AGRO modifies the classical CCG algorithm in two key ways. (1) To reduce costs resulting from over-conservative uncertainty representations, we construct an uncertainty set with known probability mass by training a VAE and projecting spherical uncertainty sets from the latent space into the space of $\boldsymbol{\xi}$. (2) Rather than attempt to solve the resulting adversarial subproblem with an embedded neural network as a mixed-binary bilinear program, we propose a PGA heuristic that quickly identifies cost-maximizing realizations using the gradient of the recourse cost with respect to $\boldsymbol{\xi}$.

Like Dumouchelle et al. (2024); Goerigk & Kurtz (2023); Chenreddy et al. (2022), we optimize "through" a trained neural network to obtain worst-case uncertainty realizations. However, rather than approximate recourse decisions using a predictive model as previous works have done for ARO (Dumouchelle et al., 2024; Bertsimas & Kim, 2024), we *exactly* solve the recourse problem as a linear program in each iteration. Doing so also removes the computational effort required for building a dataset of recourse problem solutions – i.e., solving a large number of linear programs to obtain an accurate predictive model for recourse cost given $\boldsymbol{\xi}$ and $\boldsymbol{x}$. In what follows, we describe how AGRO first learns tighter uncertainty sets and then solves the adversarial subproblem within a CCG-like scheme.

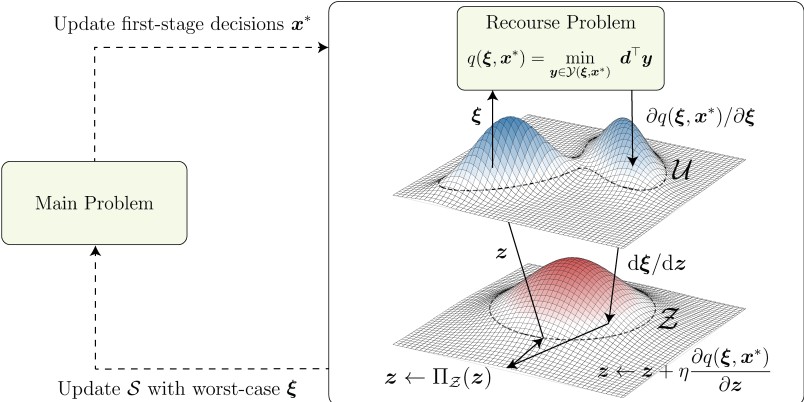

Figure 1: The AGRO solution algorithm. (i) First-stage decisions $\boldsymbol{x}^*$ are obtained by solving a main problem, which approximates the original ARO uncertainty set $\mathcal{U}$ by a finite scenario set $\mathcal{S}$ (see problem 3) (ii) The latent variable $\boldsymbol{z}$ is sampled from within $\mathcal{Z}$ and decoded to obtain an initial $\boldsymbol{\xi}$. (iii) The recourse problem is solved given $\boldsymbol{x}^*$ and $\boldsymbol{\xi}$ and its optimal objective value is differentiated with respect to $\boldsymbol{\xi}$ and (after backpropagating through the decoder $f_\theta$) $\boldsymbol{z}$. (iv) A gradient ascent step is taken to update $\boldsymbol{z}$, which is then projected onto $\mathcal{Z}$ and decoded to obtain an updated $\boldsymbol{\xi}$. Steps (iii) and (iv) are iterated until converging to a worst-case $\boldsymbol{\xi}$. (v) The worst-case $\boldsymbol{\xi}$ is then added to $\mathcal{S}$, at which point the main problem is re-optimized.

## 3.1 Deep Data-Driven Uncertainty Sets

AGRO constructs tight uncertainty sets as nonlinear and differentiable transformations of spherical uncertainty sets lying in the latent space $\mathbb{R}^L$ with $L < D$, where $D$ is the dimensionality of $\boldsymbol{\xi}$. We approach learning these transformations as a VAE estimation task. Let $\boldsymbol{z} \sim \mathcal{N}(\boldsymbol{0}, \boldsymbol{I}_L)$ be an isotropic Gaussian random variable and let $h_\phi : \mathbb{R}^D \to \mathbb{R}^L$ and $f_\theta : \mathbb{R}^L \to \mathbb{R}^D$ respectively denote the encoder and decoder of a VAE model trained to generate samples from $p_{\boldsymbol{\xi}}$ (Kingma & Welling, 2013). Specifically, $h_\phi$ is trained to map samples of uncertainty realizations to samples from $\mathcal{N}(\boldsymbol{0}, \boldsymbol{I}_L)$ while $f_\theta$ is trained to perform the reverse mapping. Here, $L$ denotes the VAE bottleneck dimension, a hyperparameter that plays a large role in determining the *fidelity* and *diversity* of generated samples (see Appendix A.3), which must be tuned through trial and error. We discuss the role of the bottleneck dimension further in Section 4.

We note that our approach can be extended to other classes of deep generative models that learn such a mapping from $\mathcal{N}(\boldsymbol{0}, \boldsymbol{I}_L)$ to $p_{\boldsymbol{\xi}}$ such as generative adversarial networks (Goodfellow et al., 2014), normalizing flows (Papamakarios et al., 2021), and diffusion models (Ho et al., 2020), which tend to produce samples with higher fidelity in general (Bond-Taylor et al., 2021). We employ a VAE architecture as VAEs are known to exhibit relatively high stability in training and low computational effort for sampling (Bond-Taylor et al., 2021), making them highly conducive to integration within an iterative optimization framework. Additionally, experimental findings suggest significant cost reductions from AGRO over classical CCG even when using a generative model that produces samples with only moderate fidelity (see Section 4).

To construct the uncertainty set $\mathcal{U}$, we first consider a "latent" uncertainty set given by the $L$-ball $\mathcal{Z} \coloneqq \mathcal{B}(\boldsymbol{0}, \Gamma)$. We select the radius $\Gamma$ such that the decoded uncertainty set $\mathcal{U} = \{f_\theta(\boldsymbol{z}) \mid \boldsymbol{z} \in \mathcal{Z}\}$ satisfies $\mathbb{P}_{\boldsymbol{\xi} \sim p_{\boldsymbol{\xi}}}(\boldsymbol{\xi} \in \mathcal{U}) \geq \alpha$ with confidence level $\delta$. Letting $\boldsymbol{\Xi} \in \mathbb{R}^{N \times D}$ be our training dataset, we leverage Theorem 1 from Hong et al. (2021) to obtain $\Gamma$ according to the following steps.

1. Split $\boldsymbol{\Xi}$ into two disjoint sets: one with $N_1 \geq \log \delta / \log \alpha$ samples reserved for calibrating the size of the uncertainty set (i.e., steps 2 and 3) and another with $(N - N_1)$ samples for training the VAE.

2. After training the VAE, project the calibration data into the latent space and sort the resulting $L^2$ norms, i.e. $r_j \coloneqq \|h_\phi(\boldsymbol{\xi}^{(j)})\|_2$ for all $\boldsymbol{\xi}^{(j)}$ in the calibration set, to obtain the order statistics of $\{r_1, \ldots, r_{N_1}\}$. We denote these order statistics by $r_{(1)} \leq \cdots \leq r_{(N_1)}$.

3. Choose $\Gamma = r_{(\ell)}$, where $j$ is defined as:

$$\ell = \min_{1 \le j \le N_1} \left\{ j \;\Big|\; \sum_{k=0}^{i-1} \binom{N_1}{k} \alpha^k (1-\alpha)^{N_1-k} \ge 1 - \delta \right\}.$$

This procedure effectively prescribes $\Gamma$ to be a robust (with respect to the sample size) $\alpha$-quantile estimate of the norm of encoded samples drawn from $p_{\boldsymbol{\xi}}$ with confidence level $\delta$. Figure 2 shows a comparison of classical uncertainty sets against a tighter uncertainty set constructed by the proposed VAE-based method for samples from a nonunimodal bivariate distribution.

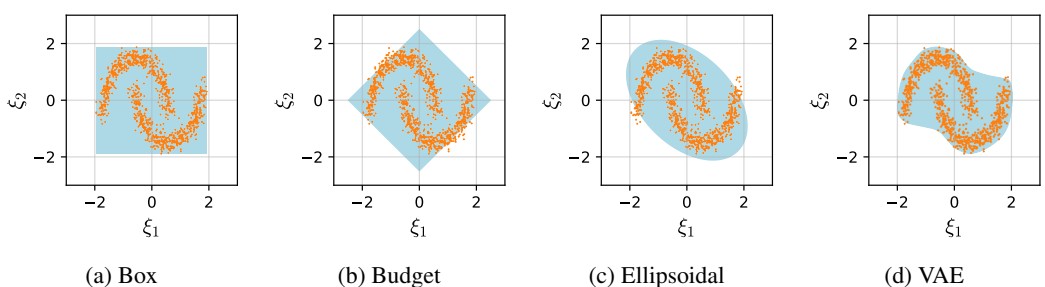

| (a) Box | (b) Budget | (c) Ellipsoidal | (d) VAE |

Figure 2: A comparison of classical uncertainty sets (a)-(c) with the proposed VAE-learned uncertainty set (d) for an illustrative bivariate distribution. By covering a smaller region in $\mathbb{R}^D$, the VAE-learned uncertainty set (d) is more likely to yield a tighter approximation of the probabilistic constraint in equation 2 and less conservative first-stage decisions.

## 3.2 Adversarial Subproblem

**Bilinear Formulation.** As the main problem in AGRO is the same as problem 3 for the classical CCG algorithm, we focus our discussion on the adversarial subproblem. Given a VAE decoder with ReLU activations $f_\theta : \mathbb{R}^L \to \mathbb{R}^D$, a latent uncertainty set $\mathcal{Z} = \mathcal{B}(\boldsymbol{0}, \Gamma)$, and the first-stage decisions $\boldsymbol{x}^*$, we formulate the adversarial subproblem as

$$\max_{\boldsymbol{z},\, \boldsymbol{\xi}} \left\{ \min_{\boldsymbol{y} \in \mathcal{Y}(\boldsymbol{\xi}, \boldsymbol{x}^*)} \boldsymbol{d}(\boldsymbol{\xi})^\top \boldsymbol{y} \;\Big|\; \boldsymbol{\xi} = f_\theta(\boldsymbol{z}),\ \boldsymbol{z} \in \mathcal{Z} \right\}. \tag{5}$$

To solve problem 5, one can reformulate it as a single-level maximization problem by taking the dual of the recourse problem and maximizing the objective with respect to $\boldsymbol{z}$ subject to (1) optimality conditions (i.e., primal/dual feasibility and strong duality) of the recourse problem and (2) constraints encoding the input-output mapping for the ReLU decoder network, $\boldsymbol{\xi} = f_\theta(\boldsymbol{z})$ (Fischetti & Jo, 2018). This formulation is presented in full as problem 7 in Appendix A.2. As discussed in Section 2.2, however, this problem can be extremely computationally demanding to solve. Therefore, we propose an alternative approach for solving problem 5 based on PGA to efficiently obtain approximate solutions.

**Projected Gradient Ascent Heuristic.** Let $q : \mathbb{R}^D \times \mathcal{X} \to \mathbb{R}$ denote the optimal objective value of the recourse problem as a function of $\boldsymbol{\xi}$ and $\boldsymbol{x}$. It is then possible to differentiate $q$ with respect to $\boldsymbol{\xi}$ by implicitly differentiating the optimality conditions of the recourse problem at an optimal solution (Amos & Kolter, 2017). Doing so provides an ascent direction for maximizing $q$ with respect to $\boldsymbol{z}$ by backpropagating through $f_\theta$. As such, we maximize $q$ using PGA over $\mathcal{Z}$.

Specifically, to solve the adversarial subproblem using PGA, we first randomly sample an initial $\boldsymbol{z}$ from a distribution supported on $\mathcal{Z}$. We then transform $\boldsymbol{z}$ to obtain the uncertainty realization $\boldsymbol{\xi} = f_\theta(\boldsymbol{z})$ and solve the recourse problem with respect to $\boldsymbol{\xi}$ to obtain $q(\boldsymbol{\xi}, \boldsymbol{x}^*)$ and $\partial q(\boldsymbol{\xi}, \boldsymbol{x}^*)/\partial \boldsymbol{\xi}$. Using automatic differentiation, we obtain the gradient of $\boldsymbol{\xi}$ with respect to $\boldsymbol{z}$ and perform the update:

$$\boldsymbol{z} \leftarrow \Pi_{\mathcal{Z}} \left( \boldsymbol{z} + \eta \frac{\partial q(\boldsymbol{\xi}, \boldsymbol{x}^*)}{\partial \boldsymbol{\xi}} \frac{\mathrm{d}\boldsymbol{\xi}}{\mathrm{d}\boldsymbol{z}} \right),$$

where $\eta > 0$ denotes the step-size hyperparameter and $\Pi_{\mathcal{Z}}$ denotes the projection operator given by $\Pi_{\mathcal{Z}}(\boldsymbol{z}) = \min\{\Gamma, \|\boldsymbol{z}\|_2\} \times \boldsymbol{z}/\|\boldsymbol{z}\|_2$. By projecting onto the set $\mathcal{Z}$ with radius $\Gamma$, we ensure

that the solution to the adversarial subproblem is neither unrealistic nor overly adversarial. This procedure is repeated until convergence of $q(f_\theta(z), x^*)$ to a local maximum or another criterion (e.g., a maximum number of iterations) is met.

PGA is not guaranteed to converge to a worst-case uncertainty realization (i.e., global maximum of problem 5) as $q(f_\theta(z), x^*)$ is nonconcave in $z$. To escape local minima, we randomly initialize PGA with $I$ different samples of $z$ in each iteration and select the worst-case uncertainty realization obtained after maximizing with PGA. We then update the set $S$ and re-optimize the main problem to complete one iteration of AGRO. Additional iterations of AGRO are then performed until convergence of the main problem and adversarial subproblem objectives, i.e., $|\gamma - q(f_\theta(z), x^*)| < \epsilon$, for tolerance $\epsilon$. Algorithm 2 provides the full pseudocode for AGRO.

# 4 Experiments

To demonstrate AGRO's efficacy in reducing planning costs, we apply our approach in two sets of experiments: a synthetic production-distribution problem and a real-world power system capacity expansion problem. For all experiments, we let $\alpha = 0.95$ and fix the confidence level $\delta$ to be $0.05$ (see Section 3.1).

## 4.1 Production-Distribution Problem

**Problem Description.** We first apply AGRO to an adaptive robust production-distribution problem with synthetic demand data in order to evaluate its performance over a range of uncertainty distributions. We consider a two-stage problem in which the first-stage decisions $x \in \mathbb{R}^{|\mathcal{I}|}$ correspond to the total number of items produced at each facility $i \in \mathcal{I}$ while second-stage decisions correspond to the quantities $y \in \mathbb{R}^{|\mathcal{I}| \times |\mathcal{J}|}$ distributed to a set of demand buses $\mathcal{J}$ after demands $\xi \in \mathbb{R}^{|\mathcal{J}|}$ have been realized. Production incurs a unit cost of $c$ in the first stage while shipping and unmet demand incur unit costs of $d \in \mathbb{R}^{|\mathcal{I}| \times |\mathcal{J}|}$ and $d' \in \mathbb{R}$ respectively. Here, transportation costs vary by origin and destination while the cost of unmet demand is uniform across buses.

**Experimental Setup.** In each experiment, we generate the demand dataset $\Xi \in \mathbb{R}^{N \times |\mathcal{J}|}$ by drawing samples from a Gaussian mixture distribution with 3 correlated components. We vary the problem size over the range $(|\mathcal{I}|, |\mathcal{J}|) \in \{(4, 3), (8, 6), (12, 9), (16, 12)\}$, and perform 50 experimental trials for each problem size. In each trial, we sample a dataset of 2500 samples, each of which is then divided into a 1000-sample VAE training set (split $80/20$ for model training/validation), 500-sample $\Gamma$-calibration set, and 1000-sample test set. Additional implementation details are provided in Appendix B.1.

Towards understanding how the VAE bottleneck dimension informs optimization outcomes, we vary $L \in \{1, 2, 4\}$. We report standard generative fidelity and diversity metrics for trained VAEs in Appendix B.1.1. To evaluate costs, we fix the solved investment decision $x$ and compute $c^\top x + \hat{F}^{-1}(\alpha; x)$, where $\hat{F}^{-1}(\alpha; x)$ is the $\alpha$-quantile of recourse costs obtained over the test set given $x$. We compare cost and runtime results obtained by AGRO to those obtained by a classical CCG algorithm, for which we consider both budget and ellipsoidal uncertainty sets with sizes calibrated according to the approach described by Hong et al. (2021).

**Costs and Runtime.** The relative improvement in out-of-sample costs (i.e., $c^\top x + \hat{F}^{-1}(\alpha; x)$) obtained by AGRO in comparison to CCG is shown in Figure 3. Cost comparisons for CCG with ellipsoidal uncertainty sets are omitted for the cases with $|\mathcal{J}| > 6$ as the resulting CCG subproblem could not be solved to optimality within the time limit of 900 seconds. Figure 3 shows that AGRO's relative advantage over classical CCG in minimizing costs grows as the dimensionality of uncertainty increases. This is expected as classical uncertainty sets are known to provide looser approximations of probabilistic constraints in higher dimensional settings (Lam & Qian, 2019). Runtime results provided in Table 2 (see Appendix B.1.1) also show that runtimes for AGRO scale much better than those for CCG with a budget uncertainty set. In particular, CCG is solved 80 times faster than AGRO with $L = 4$ for $|\mathcal{J}| = 3$ but is about 10 times slower than AGRO with $L = 4$ for $|\mathcal{J}| = 12$.

**Bottleneck Dimension.** The effect of the VAE bottleneck dimension on out-of-sample costs can be observed from the box-plot in Figure 3. In particular, solutions obtained from AGRO using VAEs with $L = 4$ outperform CCG more consistently than solutions obtained with $L < 4$. This is likely due to the fact that VAEs with $L = 4$ achieve a greater *coverage* of $p_{\boldsymbol{\xi}}$ (see Table 4 in Appendix B.1.1), and consequently, will consistently yield uncertainty sets that satisfy the probabilistic constraint. On the other hand, models with lower bottleneck dimensions are more prone to "overlooking" adversarial realizations due to inadequate coverage of $p_{\boldsymbol{\xi}}$, which causes occasional underestimation of worst-case recourse costs and under-production in the first stage. However, this effect is reversed in the low-dimensional case of $|\mathcal{J}| = 3$ as the VAEs with higher bottleneck dimensions, which score lower on generative *fidelity* metrics (see Table 3), over-produce in the first stage in anticipation of highly adversarial yet unrealistic demand realizations.

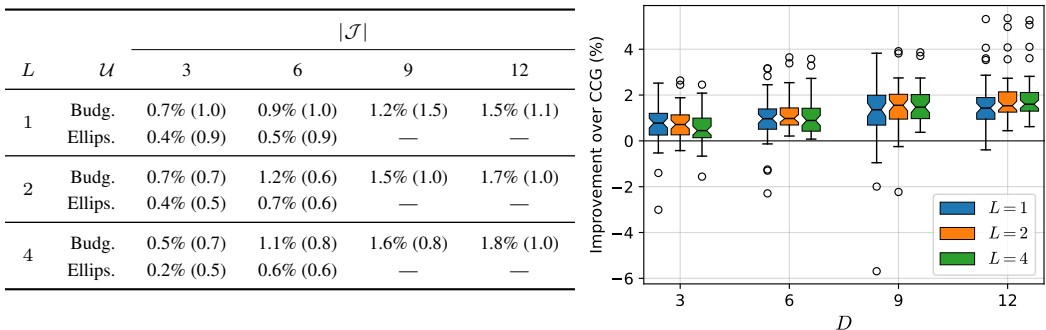

| $L$ | $\mathcal{U}$ | $\lvert\mathcal{J}\rvert$ 3 | 6 | 9 | 12 |
|---|---|---|---|---|---|
| 1 | Budg. | 0.7% (1.0) | 0.9% (1.0) | 1.2% (1.5) | 1.5% (1.1) |
| | Ellips. | 0.4% (0.9) | 0.5% (0.9) | — | — |
| 2 | Budg. | 0.7% (0.7) | 1.2% (0.6) | 1.5% (1.0) | 1.7% (1.0) |
| | Ellips. | 0.4% (0.5) | 0.7% (0.6) | — | — |
| 4 | Budg. | 0.5% (0.7) | 1.1% (0.8) | 1.6% (0.8) | 1.8% (1.0) |
| | Ellips. | 0.2% (0.5) | 0.6% (0.6) | — | — |

Figure 3: (Left): Average cost improvement with standard deviations for AGRO over CCG with budget and ellipsoidal uncertainty sets. (Right): Box-plots showing relative improvement for AGRO over CCG with budget uncertainty. The relative cost advantage of VAE-learned uncertainty sets over classical uncertainty sets increases with the dimensionality of uncertainty.

## 4.2 Capacity Expansion Planning

We now consider the problem of robust capacity expansion planning for a regional power system with historical renewable capacity factor and energy demand data. Capacity expansion models are a centerpiece for long-term planning of energy infrastructure under supply, demand, and cost uncertainties (Zhang et al., 2024; Roald et al., 2023). A number of works have adopted ARO formulations for energy systems planning with a particular focus on ensuring system reliability under mass adoption of renewable energy with intermittent and uncertain supply (Ruiz & Conejo, 2015; Amjady et al., 2017; Minguez et al., 2017; Abdin et al., 2022). In models with high spatiotemporal resolutions, supply and demand uncertainties are complex, high-dimensional, and nonunimodal, exhibiting nonlinear correlations such as temporal autocorrelations and spatial dependencies. These patterns cannot be accurately captured by classical uncertainty sets. As a result, tightening uncertainty representations for robust capacity expansion planning has the potential to significantly lower investment costs over conventional ARO methods while still maintaining low operational costs day-to-day.

**Problem Description.** We use AGRO to solve a two-stage adaptive robust generation and transmission expansion planning problem for the New England transmission system. The model we consider most closely follows that of (Minguez et al., 2017) and has two stages: an initial investment stage and a subsequent economic dispatch (i.e., recourse) stage. The objective function is a weighted sum of investment costs plus annualized worst-case economic dispatch costs. We let $\mathcal{N}$, $\mathcal{T}$, and $\mathcal{P}$ denote the set of buses, hourly operational periods, and renewable energy sources respectively; here, $|\mathcal{N}| = 6$, $|\mathcal{T}| = 24$, and $|\mathcal{P}| = 3$. Investment decisions for this problem include installed capacities for (1) dispatchable, solar, onshore wind, and offshore wind power plants for all buses, (2) battery storage at each bus, and (3) transmission capacity for all edges in $\mathcal{N} \times \mathcal{N}$. The economic dispatch stage occurs after the realization of demand and renewable capacity factors, $\boldsymbol{\xi} \in \mathbb{R}^{|\mathcal{N}||\mathcal{T}||\mathcal{P}|}$. This stage determines optimal hourly generation, power flow, and storage decisions to minimize the combined cost of load shedding and variable costs for dispatchable generation; these decisions are subject to ramping, storage, and flow balance constraints. The full formulations for both stages are presented in Appendix B.2.

**Experimental Setup.** After processing features (see Appendix B.2), we have a dataset with $N = 7300$ samples and $D = 349$ distinct features. Eight experimental trials are performed for AGRO with each trial having 6200 VAE training samples (split 80/20 for model training/validation), 100 $\Gamma$-calibration samples, and 1000 test samples (all sampled uniformly at random). Similarly to Section 4.1, we evaluate costs by fixing solved investment decisions and computing the $\alpha$-quantile of economic dispatch costs over the test set. We consider a ball-box latent uncertainty set for AGRO, which intersects the latent uncertainty set described in Section 3.1 with a box uncertainty set in the latent space (i.e., $\min_i h_\phi(\Xi_{i,:})_j \leq z_j \leq \max_i h_\phi(\Xi_{i,:})_j$ for $j = 1, \ldots, L$). We compare AGRO against a classical CCG algorithm using a budget-box uncertainty set, which is commonly used in the power systems literature (Bertsimas et al., 2012; Ruiz & Conejo, 2015; Amjady et al., 2017; Abdin et al., 2022). Here, we calibrate $\mathcal{U}$ similarly to $\Gamma$ using 100 samples (see Section 3.1). We do not report results obtained using an ellipsoidal uncertainty set as the corresponding CCG subproblem could not be solved to optimality within the time limit of 900 seconds. As was the case in Section 4.1, we report results for three choices of the VAE bottleneck dimension: $L \in \{2, 4, 8\}$.

|  | Value-at-Risk | | Upper Bound | | Error | | Total | | Train | | Subproblem | |
|---|---|---|---|---|---|---|---|---|---|---|---|---|
| CCG | 11.3 | (0.1) | 15.3 | — | 1088% | (1935) | 29183 | — | — | — | 1263 | (937) |
| $L = 2$ | 14.9 | (3.5) | 11.2 | (1.1) | **14%** | (52) | **2162** | (2128) | 207 | (5) | 239 | (796) |
| $L = 4$ | **10.4** | (0.3) | 12.8 | (0.2) | 114% | (64) | 2299 | (509) | 203 | (2) | 267 | (225) |
| $L = 8$ | 10.8 | (0.3) | 13.6 | (0.5) | 112% | (40) | 3070 | (618) | 205 | (1) | 347 | (253) |

Table 1: AGRO and CCG cost and runtime performance estimates averaged over all trials (standard deviations shown in parentheses). Given the solved first-stage decisions $\boldsymbol{x}^*$, *Value-at-Risk* denotes the sample-estimated objective value, $\boldsymbol{c}^\top \boldsymbol{x}^* + \hat{F}^{-1}(\alpha; \boldsymbol{x})$, while *Upper Bound* denotes the CCG/AGRO upper bound, $\boldsymbol{c}^\top \boldsymbol{x}^* + \gamma^*$ (both in units of $ billions). *Error* denotes the mean absolute percent error of worst-case recourse costs obtained by the subproblem compared to an out-of-sample VaR estimate of recourse costs averaged over all iterations and folds. *Total*, *Train*, and *Subproblem* respectively denote average total solve time (not including training), average training time, and average subproblem runtime in seconds.

**Costs and Runtime.** Table 1 reports cost, runtime, and VaR approximation error results for our AGRO and CCG implementations. For $L = 4$ and $L = 8$, AGRO obtains investment decisions that yield lower total costs compared to CCG as evaluated over held-out samples. In particular, costs are minimized by AGRO with $L = 4$, which yields an 8% average reduction in total costs over CCG. For all $L$, the average total runtime for AGRO, which includes VAE training time, is also observed to be substantially lower than that of CCG. This is largely owing to the computational difficulty of solving the bilinear subproblem for CCG.

**Approximation of VaR.** To empirically compare the tightness of VAE-learned and classical uncertainty sets with respect to approximating VaR, we compare the relative error of worst-case recourse costs obtained by the adversarial subproblems (i.e., $q(\boldsymbol{\xi}^i, \boldsymbol{x}^*)$) with out-of-sample VaR estimates (i.e., $\hat{F}^{-1}(\alpha; \boldsymbol{x}^*)$) for all iterations. Table 1 shows that worst-case recourse costs obtained by AGRO with $L = 2$ most closely approximate the true VaR. We visualize this phenomenon by plotting worst-case recourse costs against VaR estimates obtained in all iterations in Figure 5 (see Appendix B.2.1). Figure 5 shows that AGRO almost always *overestimates* VaR for $L = 4$ and $L = 8$, which empirically suggests that the VAE-learned uncertainty sets provide a reliable approximation for the probabilistic constraint in equation 2. The VAEs trained with $L = 2$ – despite yielding the lowest error in approximating VaR – achieve a much lower coverage of $p_{\boldsymbol{\xi}}$ than other models (see Table 5 in Appendix B.2.1). As a result, the effective coverage of $\mathcal{U}$ is less than $\alpha$, and $\mathcal{U}$ is no longer a safe approximation for the chance constraint. Consequently, AGRO is unable to produce sufficiently adversarial realizations, and by extension, achieve low planning costs.

**Realism of Generated Contingencies.** We visualize AGRO's advantage over CCG in avoiding unrealistic contingencies by plotting solved worst-case realizations for CCG and AGRO in Figure 4. We observe that CCG generates highly unrealistic realizations in which capacity factors drop sharply to zero before immediately rising to their nominal values. Importantly, weather-related correlations between variables are not maintained; most notably, temporal autocorrelation (i.e., smoothness) and spatial correlation of both load and capacity factors across the system. Accounting for these features during investment planning is crucial for effectively leveraging the complementarities between generation, storage, and transmission. On the other hand, AGRO is able to identify uncertainty realizations that are simultaneously cost-maximizing and realistic when utilizing a sufficiently high-

fidelity VAE (i.e., with $L = 4$). This can be observed from the realistic temporal autocorrelation and spatial correlations demonstrated by the AGRO-generated realizations.

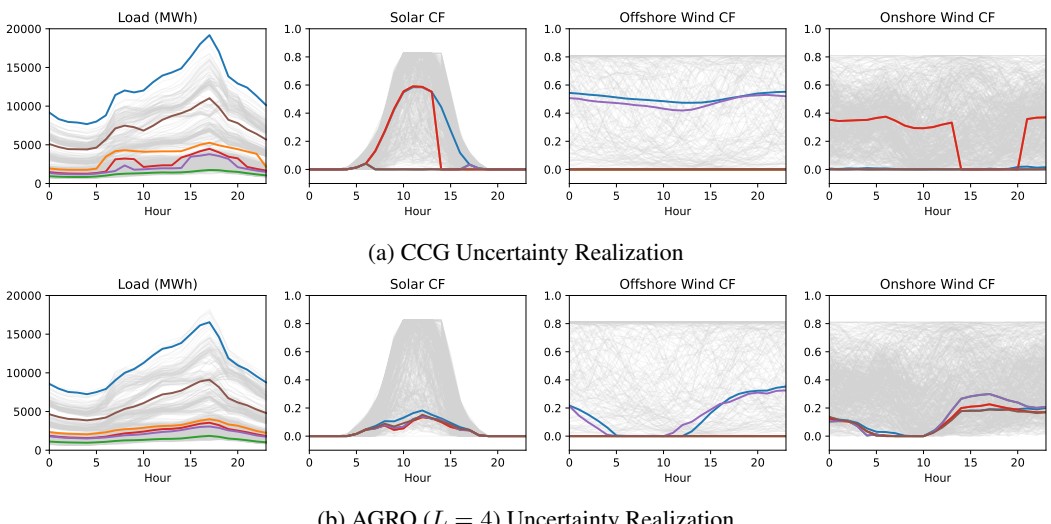

(a) CCG Uncertainty Realization

(b) AGRO ($L = 4$) Uncertainty Realization

Figure 4: Comparison of worst-case uncertainty realizations as obtained by (a) CCG and (b) AGRO with $L = 4$. Different colors are used to represent each of the six buses in the power system. A random sample of 100 observations is shown in gray to represent the historical distribution of load and capacity factors.

**Bottleneck Dimension.** As in the production-distribution study, we observe that the optimal bottleneck dimension for the capacity expansion study ($L = 4$) does not necessarily yield the best generative model performance metrics (see Table 5 in Appendix B.2.1). This stems from the conservative nature of uncertainty set-based approximations of chance constraints, which aim to cover $\alpha$ probability mass rather than the region containing the least adversarial realizations, which would provide the tightest approximation. VAEs with reduced fidelity or diversity performance mitigate resource over-commitment in the first stage by avoiding overly adversarial scenarios. However, overly low fidelity or diversity (e.g., $L = 2$) leads AGRO to underestimate realistic contingencies, resulting in systemic under-commitment and higher recourse costs.

The effectiveness of AGRO's decision-making ultimately depends on the quality of the underlying VAE as a generative model. To prevent systemic underestimation of recourse costs, it is essential that one first select $L$ large enough to ensure that generated samples are both diverse and realistic. This can be assessed through visual inspection and validated using quantitative metrics such as density and coverage. Reducing $L$ to narrow the uncertainty set below $\alpha$ should only be considered when enough data is available to provide robust out-of-sample estimates of recourse costs, ensuring a reliable approximation of the chance constraint.

# 5 Conclusion

The increasing availability of data for large-scale system operations has fueled interest in data-driven optimization for planning. This work introduces AGRO, a novel approach to two-stage ARO that integrates a VAE within an optimization framework to generate realistic and adversarial representations of operational uncertainties. Building on the classical CCG algorithm, AGRO leverages differentiable optimization techniques to identify cost-maximizing uncertainty realizations within a VAE-learned uncertainty set. Empirical results demonstrate that AGRO outperforms standard ARO methods by more tightly approximating value-at-risk constraints on recourse costs, achieving up to a 1.8% reduction in total costs in a synthetic production-distribution problem and up to an 8% reduction in a regional power system expansion problem using historical energy supply and demand data.

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

# A    Methodological Details

## A.1    Uncertainty Sets

The box, budget, and ellipsoidal uncertainty sets are constructed in a data-driven setting as

$$\mathcal{U}^{\text{box}} = \left\{ \boldsymbol{\xi} \in \mathbb{R}^D \ \mid \ \hat{\boldsymbol{\xi}}^{\text{min}} \leq \boldsymbol{\xi} \leq \hat{\boldsymbol{\xi}}^{\text{max}} \right\} \tag{6a}$$

$$\mathcal{U}^{\text{budget}} = \left\{ \boldsymbol{\xi} \in \mathbb{R}^D \ \mid \ \sum_{i=1}^{D} \hat{\Sigma}_{ii}^{-1} |\boldsymbol{\xi}_i - \hat{\mu}_i| \leq \Gamma_{\text{budget}} \right\} \tag{6b}$$

$$\mathcal{U}^{\text{ellipse}} = \left\{ \boldsymbol{\xi} \in \mathbb{R}^D \ \mid \ (\boldsymbol{\xi} - \hat{\boldsymbol{\mu}})^\top \hat{\boldsymbol{\Sigma}}^{-1} (\boldsymbol{\xi} - \hat{\boldsymbol{\mu}}) \leq \Gamma_{\text{ellipse}} \right\}. \tag{6c}$$

Here, $\hat{\boldsymbol{\mu}}$, $\hat{\boldsymbol{\Sigma}}$, $\hat{\boldsymbol{\xi}}^{\text{min}}$, and $\hat{\boldsymbol{\xi}}^{\text{max}}$ denote the empirical mean, covariance, minimum value, and maximum value of $\boldsymbol{\xi}$ observed from the dataset while the $\Gamma$ parameters denote the uncertainty "budgets."

## A.2    Bilevel Formulation for AGRO

Letting $\boldsymbol{\pi}$ denote the dual variables of the recourse problem, the max-min problem 5 can be reformulated as the following mixed-binary bilinear program:

$$\max \quad \boldsymbol{d}(\boldsymbol{\xi})^\top \boldsymbol{y} \tag{7a}$$
$$\text{s.t.} \quad \boldsymbol{B}(\boldsymbol{\xi})\boldsymbol{y} \geq \boldsymbol{b}(\boldsymbol{\xi}) - \boldsymbol{A}(\boldsymbol{\xi})\boldsymbol{x}^*, \tag{7b}$$
$$\boldsymbol{\pi}^\top \boldsymbol{B}(\boldsymbol{\xi}) \leq \boldsymbol{d}(\boldsymbol{\xi}), \tag{7c}$$
$$\boldsymbol{y}, \boldsymbol{\pi} \geq \boldsymbol{0}, \tag{7d}$$
$$\boldsymbol{\pi}^\top (\boldsymbol{b}(\boldsymbol{\xi}) - \boldsymbol{A}(\boldsymbol{\xi})\boldsymbol{x}^*) = \boldsymbol{d}(\boldsymbol{\xi})^\top \boldsymbol{y}, \tag{7e}$$
$$\|\boldsymbol{z}\|_2 \leq \Gamma, \tag{7f}$$
$$\boldsymbol{W}^{(0)}\boldsymbol{z} + \boldsymbol{b}^{(0)} = \boldsymbol{z}^{(1)} - \tilde{\boldsymbol{z}}^{(1)}, \tag{7g}$$
$$\boldsymbol{W}^{(\ell)}\boldsymbol{z}^{(\ell)} + \boldsymbol{b}^{(\ell)} = \boldsymbol{z}^{(\ell+1)} - \tilde{\boldsymbol{z}}^{(\ell+1)}, \qquad \ell \in [K-1] \tag{7h}$$
$$\boldsymbol{W}^{(K)}\boldsymbol{z}^{(K)} + \boldsymbol{b}^{(K)} = \boldsymbol{\xi}, \tag{7i}$$
$$a_j^{(\ell)} = 1 \implies z_j^{(\ell)} = 0, \qquad j \in [D_\ell],\ \ell \in [K-1] \tag{7j}$$
$$a_j^{(\ell)} = 0 \implies \tilde{z}_j^{(\ell)} = 0, \qquad j \in [D_\ell],\ \ell \in [K-1] \tag{7k}$$
$$\boldsymbol{z}^{(\ell)}, \tilde{\boldsymbol{z}}^{(\ell)} \geq \boldsymbol{0}, \qquad \ell \in [K] \tag{7l}$$
$$\boldsymbol{a}^{(\ell)} \in \{0,1\}^{D_\ell}. \qquad \ell \in [K-1] \tag{7m}$$

Here, constraints 7b and 7c respectively enforce primal and dual feasibility, constraint 7d enforces nonnegativity of the primal and dual variables, and constraint 7e enforces strong duality. Together, constraints 7b–7e provide sufficient conditions for optimality of the recourse problem. Constraint 7f enforces $\boldsymbol{z} \in \mathcal{Z}$ to bound the range of uncertainty realizations that are considered. Additionally, we apply the main result of Fischetti & Jo (2018) to embed the mapping $\boldsymbol{\xi} = f_\theta(\boldsymbol{z})$ by introducing the binary decision variables $\{\boldsymbol{a}^{(\ell)}\}_{\ell \in [K-1]}$ and real-valued decision variables $\{\boldsymbol{z}^{(\ell)}, \tilde{\boldsymbol{z}}^{(\ell)}\}_{\ell \in [K]}$ through constraints 7g–7m. Here, $D_\ell$ denotes the number of output units in layer $\ell$ while $\boldsymbol{W}^{(\ell)} \in \mathbb{R}^{D_{\ell+1} \times D_\ell}$ and $\boldsymbol{b}^{(\ell)} \in \mathbb{R}^{D_{\ell+1}}$ denote the weight matrix and offset parameters of layer $\ell$. Note that, for any $\boldsymbol{x} \in \mathcal{X}$, problem 7 is feasible as a consequence of the primal feasibility (i.e., complete recourse) and boundedness assumptions.

## A.3    Fidelity and Diversity Metrics

We quantify performance of the VAEs as generative models for $p_{\boldsymbol{\xi}}$ using standard metrics for fidelity (precision and density) and diversity (recall and coverage). Semantically, fidelity can be understood

as measuring the "realism" of generated samples while diversity can be understood as measuring the breadth of generated samples. These metrics are formally defined by Naeem et al. (2020) as follows:

$$\text{precision} = \frac{1}{M} \sum_{j=1}^{M} \mathbb{1}_{\hat{\boldsymbol{\xi}}^{(j)} \in \text{manifold}(\boldsymbol{\xi}^{(1)}, \ldots, \boldsymbol{\xi}^{(N)})}$$

$$\text{density} = \frac{1}{kM} \sum_{j=1}^{M} \sum_{i=1}^{N} \mathbb{1}_{\hat{\boldsymbol{\xi}}_j \in \mathcal{B}(\boldsymbol{\xi}^{(i)}, \text{NND}_k(\boldsymbol{\xi}^{(i)}))}$$

$$\text{recall} = \frac{1}{N} \sum_{i=1}^{N} \mathbb{1}_{\boldsymbol{\xi}_i \in \text{manifold}(\hat{\boldsymbol{\xi}}^{(1)}, \ldots, \hat{\boldsymbol{\xi}}^{(M)})}$$

$$\text{coverage} = \frac{1}{N} \sum_{i=1}^{N} \mathbb{1}_{\exists \, j \text{ s.t. } \hat{\boldsymbol{\xi}}^{(j)} \in \mathcal{B}(\boldsymbol{\xi}^{(i)}, \text{NND}_k(\boldsymbol{\xi}^{(i)}))},$$

where $\boldsymbol{\xi}$ and $\hat{\boldsymbol{\xi}}$ respectively denote real ($N$ total) and generated samples ($M$ total), $\mathbb{1}$ denotes the indicator function, $\text{NND}_k(\boldsymbol{\xi}^{(i)})$ denotes the distance from $\boldsymbol{\xi}^{(i)}$ to its $k$-th nearest neighbor (excluding itself) in the dataset $\{\boldsymbol{\xi}^{(1)}, \ldots, \boldsymbol{\xi}^{(N)}\}$, and manifolds are defined as

$$\text{manifold}(\boldsymbol{\xi}^{(1)}, \ldots, \boldsymbol{\xi}^{(N)}) = \bigcup_{i=1}^{N} \mathcal{B}(\boldsymbol{\xi}^{(i)}, \text{NND}_k(\boldsymbol{\xi}^{(i)})).$$

Precision and density quantify the portion of generated samples that are "covered" by the real samples while recall and coverage quantify the portion of real samples that are "covered" by the generated samples. All results for both sets of experiments are obtained over the held-out test datasets using $k = 5$ and with $M = N = 1000$.

## A.4 AGRO Algorithm

---

**Algorithm 2** AGRO Algorithm

---

**Require:** Decoder $g_\theta$, Confidence Level $\alpha$, Step-size $\eta$, Initializations $I$, Convergence tolerance $\epsilon$
**Ensure:** Optimal solution $x^*$
1: Initialize Scenario set $\mathcal{S} \leftarrow \emptyset$, Iteration counter $i \leftarrow 1$
2: **while** not converged **do**
3:     $x^*, \gamma \leftarrow \arg\min_{x \in \mathcal{X}, \gamma} \{f(x) + \gamma \mid f(\xi^i, x) \leq \gamma, \forall \xi^i \in \mathcal{S}\}$          $\triangleright$ Solve main problem
4:     **for** $j = 1$ to $I$ **do**          $\triangleright$ Solve adversarial subproblem
5:        Sample $z^j \sim p_s$
6:        **while** not converged **do**
7:           Decode $\xi = g_\theta(z^j)$
8:           Solve recourse problem for $f(\xi, x^*)$
9:           $z^j \leftarrow \Pi_{\mathcal{Z}} \left( z^j + \eta \frac{\partial f(\xi, x^*)}{\partial \xi} \frac{d\xi}{dz^j} \right)$
10:       **end while**
11:     **end for**
12:     $j^* \leftarrow \arg\max_{j \in \{1, \ldots, I\}} f(g_\theta(z^j), x^*)$          $\triangleright$ Select worst-case realization
13:     Set $\xi^i \leftarrow g_\theta(z^{j^*})$
14:     **if** $f(\xi^i, x^*) \leq \gamma$ **then**          $\triangleright$ Check convergence
15:       **Terminate** and return $x^*$
16:     **else**
17:       Update $\mathcal{S} \leftarrow \mathcal{S} \cup \{\xi^i\}$          $\triangleright$ Add worst-case realization to scenario set
18:       Increment $i \leftarrow i + 1$
19:     **end if**
20: **end while**
21: **return** $x^*$

---

# B Experimental Details

## B.1 Production-Distribution Problem

**Formulation.** We formulate the two-stage production-distribution problem as

$$\min_{\boldsymbol{x}} \quad \sum_{i \in \mathcal{I}} c_i x_i + \max_{\boldsymbol{\xi} \in \mathcal{U}} q(\boldsymbol{\xi}, \boldsymbol{x}) \tag{8a}$$

$$\text{s.t.} \quad \boldsymbol{x} \geq \boldsymbol{0}, \tag{8b}$$

where the recourse cost $q(\boldsymbol{\xi}, \boldsymbol{x})$ is given as the solution to the following linear program:

$$q(\boldsymbol{\xi}, \boldsymbol{x}) = \min_{\boldsymbol{y}} \quad \sum_{i \in \mathcal{I}} \sum_{j \in \mathcal{J}} d_{ij}^1 y_{ij}^1 + \sum_{j \in \mathcal{J}} d^2 y_j^2 \tag{9a}$$

$$\text{s.t.} \quad \sum_{i \in \mathcal{I}} y_{ij}^1 + y_j^2 \geq \xi_j, \qquad\qquad j \in \mathcal{J} \tag{9b}$$

$$\sum_{j \in \mathcal{J}} y_{ij} \leq p_i x_i, \qquad\qquad i \in \mathcal{I} \tag{9c}$$

$$\boldsymbol{y}^1, \boldsymbol{y}^2 \geq \boldsymbol{0}. \tag{9d}$$

This problem is adapted as a continuous relaxation (with respect to the first-stage decisions) of the facility location problem introduced by Bertsimas & Kim (2024). Here, the first-stage decisions are the real-valued decision variables $x_i$ determining the amount of items to produce at facility $i$ at a unit cost of $c_i$ for all $i \in \mathcal{I}$. After the demands $\xi_j$ for all delivery destinations $j \in \mathcal{J}$ are realized, items are shipped from facilities to delivery destinations at a unit cost of $d_{ij}$ for facility $i$ and destination $j$. These shipment quantities are captured for $i \in \mathcal{I}$ and $j \in \mathcal{J}$ by the continuous recourse decision variables $y_{ij}$. An additional penalty is incurred for any unmet demand $y_j^2$ at destination $j$ with unit cost $d^2$. Constraints 9b define $y_j^2$ as unmet demand while constraints 9c ensure the total number of items shipped from a facility $i$ does not exceed its production, $p_i x^i$, where $p_i$ is the production factor for facility $i$.

**Generating Instances.** In each trial, we randomly sample both mean and covariance parameters of the Gaussian mixture $p_{\boldsymbol{\xi}}$ as well as parameters of the production-distribution problem, $\boldsymbol{c}$, $\boldsymbol{d}$ and, $\boldsymbol{p}$ ($d'$ is fixed to 5 in all iterations). To define the Gaussian mixture distribution $p_{\boldsymbol{\xi}}$, we first sample the mixing parameters $\boldsymbol{\zeta} \in [0,1]^3$ from a symmetric Dirichlet distribution, i.e., $\boldsymbol{\zeta} \sim \text{Dir}(\boldsymbol{1})$. For each component $k = 1, \ldots, 3$, we sample the population mean $\boldsymbol{\mu}^k \sim \mathcal{N}(\boldsymbol{0}, |\mathcal{J}| \boldsymbol{I}_{|\mathcal{J}|})$ and the population covariance $\boldsymbol{\Sigma}^k$ from the Wishart distribution with $|\mathcal{J}|$ degrees of freedom and identity scale matrix $\boldsymbol{I}_{|\mathcal{J}|}$. To generate the problem parameters $\boldsymbol{c} \in \mathbb{R}^{|\mathcal{I}|}$, $\boldsymbol{d} \in \mathbb{R}^{|\mathcal{I}| \times |\mathcal{J}|}$ and, $\boldsymbol{p} \in \mathbb{R}^{|\mathcal{I}|}$, we follow the procedure described by Bertsimas & Kim (2024). Specifically, we sample $\boldsymbol{d}$ uniformly at random from the interior of the ball $\mathcal{B}(\overline{\boldsymbol{d}}, 1.5)$ where $\overline{d}_i \sim \text{Unif}(2, 22)$ for $i \in \mathcal{I}$. We then sample $p_i \sim \text{Unif}(8, 18)$ and $c_i \sim \text{Unif}(2, 4)$ for $i \in \mathcal{I}$.

**Computational Details.** Both the encoder and decoder of our VAE correspond to three-layer networks with ReLU activations and batch normalization between layers; all intermediate fully connected layers have 32 hidden units. All VAE models are trained using the Adam optimizer with a learning rate of 0.001. Additionally, we utilize a *cyclical annealing schedule* that varies the weight of the KL-divergence term for the VAE training objective (Fu et al., 2019). All training is performed on the MIT Supercloud system (Reuther et al., 2018) using an Intel Xeon Gold 6248 machine with 40 CPUs and two 32GB NVIDIA Volta V100 GPUs, which takes approximately 30 seconds for all instances. All optimization is also performed on the MIT Supercloud system (Reuther et al., 2018) using an Intel Xeon Platinum 8260 machine with 96 cores and using Gurobi 11 (Gurobi Optimization, LLC, 2023) except when solving the AGRO subproblem, in which case results are obtained using the Cvxpylayers package (Agrawal et al., 2019).

To solve the adversarial subproblem with AGRO, we perform 10 random initializations of $\boldsymbol{z}$. For each initialization, we perform normalized PGA with a learning rate of $\eta = 0.1$ until either (a) costs obtained in successive PGA steps have converged within a tolerance of $0.01\%$ or (b) 1000 PGA steps have been performed. If the worst-case realization $\boldsymbol{\xi}$ obtained over all 10 initializations does

not exceed the lower bound obtained by the most recent iteration of the master problem (i.e., $\gamma$), we perform additional initializations until either (a) a new worst-case realization is obtained or (b) 200 initializations have been performed (in which case, AGRO terminates).

### B.1.1 Experimental Results

| $|\mathcal{I}|$ | $|\mathcal{J}|$ | Budget | Ellipsoid | AGRO ($L=1$) | AGRO ($L=2$) | AGRO ($L=4$) |
|---|---|---|---|---|---|---|
| 4 | 3 | 1.13 | 1.13 | 11.93 | 46.61 | 84.77 |
| 8 | 6 | 7.1 | 371.18 | 13.8 | 45.53 | 109.69 |
| 12 | 9 | 250.09 | — | 16.29 | 73.85 | 115.08 |
| 16 | 12 | 1199.22 | — | 26.91 | 76.59 | 113.4 |

Table 2: Runtime results for production-distribution problem (not including VAE training).

| | $|J|=3$ | | $|J|=6$ | | $|J|=9$ | | $|J|=12$ | |
|---|---|---|---|---|---|---|---|---|
| $L$ | Precision | Density | Precision | Density | Precision | Density | Precision | Density |
| 1 | 0.98 | 1.08 | 0.98 | 1.43 | 0.97 | 1.94 | 0.97 | 2.65 |
| 2 | 0.98 | 1.05 | 0.95 | 1.22 | 0.93 | 1.56 | 0.93 | 2.1 |
| 4 | 0.97 | 1.0 | 0.94 | 1.09 | 0.92 | 1.36 | 0.92 | 1.78 |

Table 3: Fidelity metrics for VAEs trained using the production-distribution problem data averaged over all 50 instances.

| | $|J|=3$ | | $|J|=6$ | | $|J|=9$ | | $|J|=12$ | |
|---|---|---|---|---|---|---|---|---|
| $L$ | Recall | Coverage | Recall | Coverage | Recall | Coverage | Recall | Coverage |
| 1 | 0.08 | 0.32 | 0.01 | 0.28 | 0.01 | 0.3 | 0.0 | 0.34 |
| 2 | 0.78 | 0.83 | 0.32 | 0.66 | 0.15 | 0.65 | 0.08 | 0.67 |
| 4 | 0.96 | 0.95 | 0.89 | 0.93 | 0.61 | 0.89 | 0.37 | 0.88 |

Table 4: Diversity metrics for VAEs trained using the production-distribution problem data averaged over all 50 instances.

## B.2 Capacity Expansion Model

The capacity expansion model is given by

$$\min_{\boldsymbol{x}} \quad \sum_{i \in \mathcal{N}} c^d x_i^d + \sum_{i \in \mathcal{N}} \sum_{p \in \mathcal{P}} c^p x_i^p + \sum_{i \in \mathcal{N}} c^b x_i^b + \sum_{(i,j) \in \mathcal{E}} c^\ell x_{ij}^\ell + \lambda \max_{\boldsymbol{\xi} \in \mathcal{U}} q(\boldsymbol{\xi}, \boldsymbol{x}) \tag{10a}$$

$$\text{s.t.} \quad \sum_{i \in \mathcal{N}} x_i^p \leq \Gamma^p \qquad\qquad\qquad\qquad\qquad\qquad p \in \mathcal{P} \tag{10b}$$

$$\boldsymbol{x}^d \in \mathbb{Z}_{\geq 0}^{|\mathcal{N}|} \tag{10c}$$

$$\boldsymbol{x}^d, \boldsymbol{x}^b, \boldsymbol{x}^\ell \geq \boldsymbol{0}. \tag{10d}$$

Here the first-stage bus-level decision variables include installed dispatchable generation capacity ($x_i^d \in \mathbb{Z}_+$), installed renewable plant capacity ($x_i^p \geq 0$) for renewable technologies $p \in \mathcal{P}$, and installed battery storage capacity ($x_i^b \geq 0$). Additionally, we consider transmission capacity ($x_{ij}^\ell \geq 0$) as a first-stage decision variable for all edges $(i, j) \in \mathcal{E}$. Importantly, installed renewable plant

capacities obey a system-wide resource availability constraint. Finally, $\max_{\boldsymbol{\xi} \in \mathcal{U}} q(\boldsymbol{\xi}, \boldsymbol{x})$ denotes the worst-case recourse cost (i.e., economic dispatch cost given by equation 11), which is incurred with unit cost $\lambda$ as part of the capacity expansion objective function. We "annualize" the worst-case recourse cost by setting $\lambda = 365$. For a given investment decision $\boldsymbol{x}$ and uncertainty realization $\boldsymbol{\xi}$, the recourse cost is given by

$$\min_{\boldsymbol{y}} \sum_{t \in \mathcal{T}} \sum_{i \in \mathcal{N}} d^f y_{it}^1 + d^s y_{it}^6 \tag{11a}$$

$$\text{s.t. } y_{it}^1 + \sum_{j:(j,i) \in \mathcal{E}} y_{jit}^2 - \sum_{j:(i,j) \in \mathcal{E}} y_{ijt}^2 + y_{it}^3 + y_{it}^4 = \xi_{it}^d - \sum_{p \in \mathcal{P}} \xi_{it}^p x_i^p \qquad i \in \mathcal{N}, t \in \mathcal{T} \tag{11b}$$

$$y_{ijt}^2 \leq x_{ij}^\ell \qquad (i,j) \in \mathcal{E}, t \in \mathcal{T} \tag{11c}$$

$$-y_{ijt}^2 \leq x_{ij}^\ell \qquad (i,j) \in \mathcal{E}, t \in \mathcal{T} \tag{11d}$$

$$y_{it}^1 \leq \nu x_i^d \qquad i \in \mathcal{N}, t \in \mathcal{T} \tag{11e}$$

$$y_{it}^1 - y_{i,t-1}^1 \leq \overline{\kappa} \nu x_i^d \qquad i \in \mathcal{N}, t \in \mathcal{T} \tag{11f}$$

$$y_{i,t-1}^f - y_{it}^1 \leq -\underline{\kappa} \nu x_i^d \qquad i \in \mathcal{N}, t \in \mathcal{T} \tag{11g}$$

$$y_{it}^5 \leq x_i^b \qquad i \in \mathcal{N}, t \in \mathcal{T} \tag{11h}$$

$$y_{it}^5 - y_{i,t-1}^5 + y_{i,t-1}^4 = 0 \qquad i \in \mathcal{N}, t \in \mathcal{T} \tag{11i}$$

$$y_{it}^3 - y_{it}^6 \leq 0 \qquad i \in \mathcal{N}, t \in \mathcal{T} \tag{11j}$$

$$\boldsymbol{y}^1, \boldsymbol{y}^5, \boldsymbol{y}^6 \geq \boldsymbol{0}. \tag{11k}$$

Here, the total dispatchable generation at bus $i$ in period $t$ is denoted by $y_{it}^1$. $y_{ijt}^2$ denotes the inflow of power from bus $i$ to bus $j$ in hour $t$. $y_{it}^4$ denotes the discharge rate of the battery located at bus $i$ in hour $t$ while $y_{it}^5$ denotes its total charge. Load shedding is denoted by $y_{it}^6$, which is assumed to be the positive part of $y_{it}^3$. $\nu$ denotes the nameplate capacity of the dispatchable plants while $\overline{\kappa}$ and $\underline{\kappa}$ respectively denote the maximum ramp-up and ramp-down rates for dispatchable generation. The objective function 11a minimizes the combined cost of fuel and a carbon tax for dispatchable generation (with unit cost jointly denoted by $c^f$) as well as load shedding (with unit cost $c^s$). First-stage decisions $x_i^d$, $x_i^p$, $x_i^b$, and $x_{ij}^\ell$ parameterize the constraints of 11. Constraints 11b impose flow conservation at each bus. Specifically, the constraint balances dispatchable generation, power transmitted to/from other buses, nodal load shedding, and battery discharging with net load (i.e., demand minus total generation from renewables). Constraints 11c and 11d impose line limits on power flows. Constraints 11e impose limits on total dispatchable generation while constraints 11f and 11g limit ramping up and down of dispatchable generation. Constraints 11h limits the total state-of-charge of batteries. Constraints 11i links battery discharging or charging (taken to be the negative of discharging) rates to battery state-of-charge. Constraints 11j ensure that only the positive part of load shedding incurs a cost. To simulate inter-day dynamics of storage and ramping, we implement circular indexing, which links battery state-of-charge and dispatchable generation rates at the end of the day with those at the beginning (Jacobson et al., 2024).

**Data Processing.** To process the data for the capacity expansion planning problem, we remove components corresponding to solar capacity factors for nighttime hours and offshore wind capacity factors for landlocked buses, both of which correspond to capacity factors of zero in all observations, which gives $\boldsymbol{\xi} \in \mathbb{R}^{427}$. In implementing CCG, we also remove redundant (i.e., perfectly multicollinear) components so that the dataset of $\boldsymbol{\xi}$ observations has full column rank when defining the uncertainty set. We then re-introduce the redundant components as affine transformations of the retained components in the adversarial subproblem using equality constraints. Consequently, the effective dimensionality of $\mathcal{U}$ is 349 while the dimensionality of $\boldsymbol{\xi}$ is 427. For AGRO, we only remove components that are constant (i.e., zero) and train the VAE on the original 427-dimensional dataset.

**Computational Details.** Both the encoder and decoder of our VAE correspond to three-layer networks with LeakyReLU activations and batch normalization between layers; each fully connected layer has 64 hidden units. All VAE models are trained using the Adam optimizer with a learning rate of 0.01. As in the case for the production-distribution problem, we utilize a cyclical annealing schedule during training. All training is performed on the MIT Supercloud system (Reuther et al., 2018) using an Intel Xeon Gold 6248 machine with 40 CPUs and one 32GB NVIDIA Volta V100

GPUs. Training times range from approximately 190 to 250 seconds for all instances. All optimization is also performed on the MIT Supercloud system (Reuther et al., 2018) using an Intel Xeon Platinum 8260 machine with 96 cores and using Gurobi 11 (Gurobi Optimization, LLC, 2023) except when solving the AGRO subproblem, in which case results are obtained using the Cvxpylayers package (Agrawal et al., 2019).

To solve the adversarial subproblem with AGRO, we perform 10 random initializations of $z$. For each initialization, we perform normalized PGA with a learning rate of $\eta = 0.03$ until either (a) costs obtained in successive PGA steps have converged within a tolerance of $0.01\%$ or (b) 1000 PGA steps have been performed. If the worst-case realization $\xi$ obtained over all 10 initializations does not exceed the lower bound obtained by the most recent iteration of the master problem (i.e., $\gamma$), we perform additional initializations until either (a) a new worst-case realization is obtained or (b) 30 initializations have been performed (in which case, AGRO terminates).

### B.2.1    Experimental Results

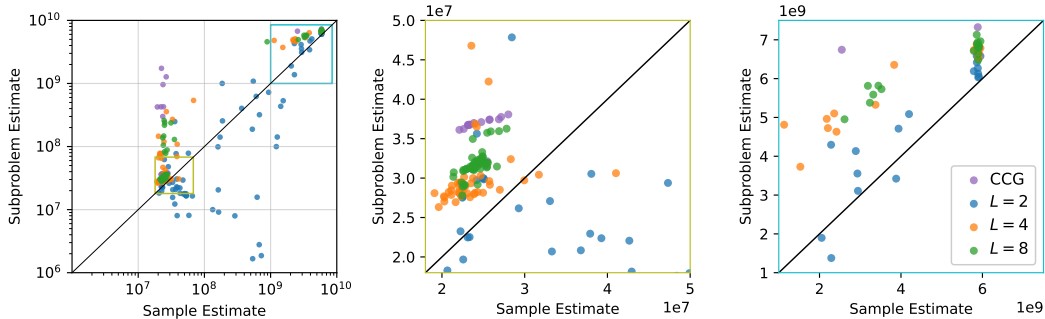

Figure 5: Comparison of sample-estimated VaR, $\hat{F}^{-1}(\alpha; x^*)$, and estimated worst-case costs obtained from solving the adversarial subproblem, $f(\xi^i, x^*)$. Each point is obtained from one iteration of AGRO/CCG. Points that are closer to the diagonal line indicate a more accurate estimate of VaR. The middle and right-hand plots provide a zoomed-in view of two regions with a high concentration of points. Subproblem estimates are almost always greater than sample estimates except in the case of AGRO with $L = 2$ (blue).

| $L$ | Precision | Density | Recall | Coverage |
|---|---|---|---|---|
| 2 | 0.84 | 1.05 | 0.11 | 0.57 |
| 4 | 0.89 | 1.45 | 0.21 | 0.77 |
| 8 | 0.96 | 1.72 | 0.39 | 0.89 |

Table 5: Fidelity and diversity metrics for VAEs trained using the capacity expansion problem dataset averaged over all eight trials.

