# OpenReview forum: "A Deep Generative Learning Approach for Two-stage Adaptive Robust Optimization"
_ICLR.cc/2025/Conference — ICLR 2025 Poster_

### Official Review · Reviewer_GwvX · 2024-10-27

**Soundness:** 2
**Presentation:** 3
**Contribution:** 3
**Rating:** 6
**Confidence:** 4

**Summary:**

This paper proposes the AGRO algorithm, which performs adversarial generation for two-stage adaptive robust optimization using a variational autoencoder. By decomposing the optimization problem to solving the 'main' problem and the adversarial subproblem iteratively, AGRO can provide tighter uncertainty estimation and lead to better optimization outputs.

**Strengths:**

1. The paper is well-written, easy to follow, and understand.

2. With VAE-learned uncertainty, the proposed AGRO method does tighten the uncertainty bounds and leads to better optimization outcomes. An intuitive example in Figure 2 and experimental results clearly demonstrate this.

**Weaknesses:**

1. In Section 3.2, the author proposes a projected gradient ascent heuristic method to optimize $q$. Although this PGA method is well-explained in the article and I understand why the author uses it, I still expect an ablation study on directly optimizing $q$ to show if PGA could still guarantee some level of optimization quality and if there are any speed improvements.

2. Although the proposed AGRO method is an improvement based on the CCG method, in the experiments section, the author should compare with more baselines for the two-step optimization problem, which I believe is a well-studied problem with many methods proposed to solve it.

**Questions:**

My main question for this work is: Is the optimization problem the author wants to solve exactly a linear optimization problem (see Eq. 1, 7, 9, 10)? If so, there are already many tools for solving linear optimization problems, so why is the proposed method better than those?

If not, what kind of optimization problem does AGRO solve? Only convex?

---

> ### Author Response · Authors · 2024-11-23
> **Comparison of PGA with other methods**
>
> We kindly refer the reviewer to our response for Reviewer vrQ2 entitled "Comparison of PGA with other methods".

---

> ### Author Response · Authors · 2024-11-23
> **Comparison of AGRO and CCG with other methods for ARO**
>
> Several methods are available for solving ARO problems, but the cost outcomes of these methods are always bounded below by those obtained with CCG, which is an exact solution method for ARO with linear recourse. This holds true for both traditional approaches, such as linear decision rules [1], and more recent machine learning-assisted ARO techniques [2, 3], both of which approximate recourse decisions as functions of uncertain parameters in order to reduce computational runtime at the cost of optimality. Other exact solution methods, such as the cut-generation method proposed in [4], are also applicable but ultimately produce the same planning outcomes as CCG.
>
> To be concrete, AGRO is contrasted with previous works on learning-assisted ML [2,3] in that AGRO does not use an ML model to predict recourse costs and instead exactly computes the the recourse problem in each iteration. While this restricts AGRO's applicability to problems with linear recourse (in contrast to [2], which can address more general recourse problems involving mixed-integer variables), it offers a significant advantage. AGRO operates as an exact solution method that only requires training on a dataset of realizations, bypassing the need to generate datasets of realizations paired with potentially costly-to-compute recourse costs.
>
> Ultimately, the core advantage of our approach lies in its ability to reduce costs by avoiding overly conservative planning associated with loose uncertainty sets. For these reasons, we do not compare our method with other ARO solution techniques, as these methods cannot outperform CCG in terms of minimizing planning costs.
>
> References:
>
> [1] Kuhn, Daniel, Wolfram Wiesemann, and Angelos Georghiou. "Primal and dual linear decision rules in stochastic and robust optimization." Mathematical Programming 130 (2011): 177-209.
>
> [2] Dumouchelle, Justin, et al. "Neur2RO: Neural two-stage robust optimization." The Twelfth International Conference on Learning Representations. 2023.
>
> [3] Thiele, Aurélie, Tara Terry, and Marina Epelman. "Robust linear optimization with recourse." Rapport technique (2009): 4-37.

---

> ### Author Response · Authors · 2024-11-23
> **Question regarding applicability of AGRO**
>
> AGRO is designed for two-stage adaptive robust optimization (ARO) problems with linear recourse, where $\min_{y \in \mathcal{Y}(\boldsymbol{x}, \boldsymbol{\xi})} \boldsymbol{d}(\boldsymbol{\xi})^\top \boldsymbol{y}$ is a linear optimization problem. Two-stage ARO is generally nonconvex due to its min-max-min structure. Specifically, while the innermost problem $\min_{y \in \mathcal{Y}(\boldsymbol{x}, \boldsymbol{\xi})} \boldsymbol{d}(\boldsymbol{\xi})^\top \boldsymbol{y}$ is a linear program, its optimal objective value is convex in $\boldsymbol{\xi}$ [1]. This implies that the inner max-min problem is a convex maximization problem, which is generally challenging to solve. Consequently, one must rely on iterative solution algorithms, such as column-and-constraint generation (CCG) and cut-generation algorithms [2], or simplifying approaches, such as linear decision rules [3], which reformulate the original nonconvex problem into a tractable single-stage problem.
>
> Although we do not discuss this extension in the submission, AGRO can also be used to solve nonlinear single-stage robust optimization problems of the form:
> \begin{align*}
>     \min_{\boldsymbol{x} \in \mathcal{X}} \quad & \boldsymbol{c}^\top \boldsymbol{x} \quad \mathrm{s.t.} \quad f(\boldsymbol{x}, \boldsymbol{\xi}) \leq \boldsymbol{0}, \ \forall \boldsymbol{\xi} \in \mathcal{U},
> \end{align*}
> where $f$ is convex in $\boldsymbol{x}$ and differentiable in $\boldsymbol{\xi}$. This generalization can be achieved using a constraint generation approach, similar to CCG. In this method, AGRO generates adversarial realizations $\boldsymbol{\xi}$ by maximizing $f(\boldsymbol{x}, \boldsymbol{\xi})$ for a given $\boldsymbol{x}$, which are then added to a finite scenario set.
>
> Building on its ability to address the challenges of two-stage ARO, AGRO’s core methodology also offers a promising foundation for broader applications in robust optimization. By leveraging deep unsupervised learning to efficiently identify adversarial scenarios, AGRO contributes a novel framework for tackling high-dimensional uncertainty in optimization. While this work emphasizes two-stage ARO, we envision future efforts extending AGRO's generative framework to new optimization paradigms. This includes advancing risk estimation for large-scale systems and identifying out-of-sample or out-of-distribution contingencies. These applications highlight AGRO’s potential to drive innovation in uncertainty-aware optimization through the integration of generative methods and robust optimization principles.
>
> [1] Bertsimas, Dimitris, et al. "Adaptive robust optimization for the security constrained unit commitment problem." IEEE transactions on power systems 28.1 (2012): 52-63.
>
> [2] Thiele, Aurélie, Tara Terry, and Marina Epelman. "Robust linear optimization with recourse." Rapport technique (2009): 4-37.
>
> [3] Kuhn, Daniel, Wolfram Wiesemann, and Angelos Georghiou. "Primal and dual linear decision rules in stochastic and robust optimization." Mathematical Programming 130 (2011): 177-209.

---

### Official Review · Reviewer_A31W · 2024-11-05

**Soundness:** 3
**Presentation:** 3
**Contribution:** 2
**Rating:** 6
**Confidence:** 3

**Summary:**

This paper addresses the two-stage adaptive robust optimization (ARO) problem, where a key challenge is constructing an effective uncertainty set. The authors propose using a deep generative model to learn the uncertainty set, aiming to avoid overly conservative optimization. The method is evaluated on a synthetic production-distribution problem and a regional power system expansion problem.

**Strengths:**

1. The paper is well-written and clear.

2. Leveraging deep generative models to learn the uncertainty set is a promising approach.

**Weaknesses:**

1. The Projected Gradient Ascent (PGA) method does not guarantee convergence to the worst-case uncertainty realization. Although the authors propose randomly initializing PGA with different samples of $z$ for empirical performance, providing some theoretical analysis on the approximation error would be beneficial.

2. The performance improvement of the proposed method is minimal.

**Questions:**

1. The paper suggests that the framework is general and could also be applied to diffusion models. However, diffusion model training involves matching the score of the noised distribution, and samples cannot be easily obtained during training. Could you elaborate on how diffusion models would integrate with your proposed framework?

2. How do you ensure the uncertainty set learned by the VAE is sufficiently tight? When optimizing over the latent space, are there any constraints? If not, is there a risk that the algorithm could select an overly conservative worst-case realization?

---

> ### Author Response · Authors · 2024-11-23
> **Theoretical guarantees for PGA**
>
> Thank you for the reviewer’s thoughtful feedback on performance guarantees. Our primary objective was to develop a learning-based approach to reduce planning costs associated with overly conservative strategies that resulting from "loose" uncertainty sets (e.g., polyhedral, elliptical, etc.). To achieve this, we proposed a learning and optimization framework that generates nonconvex uncertainty sets, recognizing that this comes at the expense of theoretical guarantees due to the bilinear (and inherently nonconvex) nature of the resulting subproblem.
>
> Providing theoretical guarantees for approximation error is inherently challenging, as determining the value of the global minimum -- let alone the optimal solution -- for a nonconvex optimization problem is a provably NP-hard task [1]. Given these limitations, we rely on experimental validation to demonstrate the effectiveness of AGRO. Our results underscore the practical advantages of this approach in greatly reducing costs for the widely studied problem of long-term energy system planning with real-world supply and demand data.
>
> For completeness, we are generating additional results from applying Gurobi's nonconvex optimizer to solve the adversarial subproblem for the production distribution problem, which is considerably lower dimensional (and perhaps more tractable) than the capacity expansion problem. The final submission will include findings regarding the optimality of solutions obtained with PGA compared to provably optimal solutions, should they be achievable. In the case that provably optimal solutions are not achievable as was the case for the capacity expansion study (see our response to Reviewer vrQ2 entitled"Comparison of PGA with other methods"), the final submission will include a brief summary of these findings.
>
> References:
>
> [1] Danilova, Marina, et al. "Recent theoretical advances in non-convex optimization." High-Dimensional Optimization and Probability: With a View Towards Data Science. Cham: Springer International Publishing, 2022. 79-163.

---

> ### Author Response · Authors · 2024-11-23
> **Performance improvement of AGRO**
>
> Our computational experiments show promising performance of AGRO as we achieve up to 10\% reduction in total costs, which is considered substantial in the context of regional energy system planning; in our capacity expansion study, a 10\% savings amounts to billions of dollars (see Table 1).

---

> ### Author Response · Authors · 2024-11-23
> **Applicability of diffusion models**
>
> We thank the reviewer for raising this point. To clarify, our proposed framework does not involve generating samples during the training phase of the generative model. Instead, the generative model is first trained on the dataset of uncertainty realizations and then used to generate adversarial realizations during the optimization phase. During this phase, the only requirement is that the generated adversarial realizations be differentiable with respect to the inputs of the generative model. As such, we believe diffusion models can be integrated into the AGRO framework.
>
> In the revised version of the manuscript, we explicitly clarify that the optimization phase operates on fully trained generative models and does not assume sample generation during their training.

---

> ### Author Response · Authors · 2024-11-23
> **Tightness of AGRO uncertainty sets**
>
> While we do not provide formal theoretical guarantees, we believe that the uncertainty sets learned by AGRO are naturally tight due to the design of the generative model. For VAEs in particular, the training process penalizes reconstruction error, which discourages the generation of unrealistic samples and ensures that reconstructed outputs align with the typical set of the target distribution. In our experiments, we observe this behavior through (1) out-of-sample evaluations of planning costs, where tighter uncertainty sets result in lower costs while still satisfying the chance constraint, and (2) visual comparisons of worst-case realizations from AGRO and CCG, which show that AGRO produces significantly more realistic adversarial scenarios.
>
> To address the concern about overly conservative worst-case realizations, we explicitly constrain the optimization over the latent space to $\boldsymbol{z} \in \mathcal{Z}$ (see Figure 1). This constraint ensures that the selected adversarial scenarios remain realistic and prevents the algorithm from yielding overly conservative solutions.

---

> ### Comment · Reviewer_A31W · 2024-12-02
>
> Thank you for your response. As the paper primarily relies on empirical performance without providing any theoretical guarantees and is evaluated on only two datasets, I believe it is a borderline paper and will maintain my score.

---

### Official Review · Reviewer_foZ7 · 2024-11-07

**Soundness:** 3
**Presentation:** 3
**Contribution:** 3
**Rating:** 6
**Confidence:** 2

**Summary:**

This paper presents a novel deep generative approach using Variational Autoencoders (VAE) to tackle two-stage adaptive robust optimization (ARO) under high-dimensional uncertainty. Traditional ARO approaches for constructing the uncertainty set $\mathcal{U}$ tend to be overly conservative, often leading to excessive resource allocation in scenarios with high-dimensional and irregularly distributed uncertainties. The proposed AGRO method mitigates this issue by incorporating VAE embedding and column-and-constraint generation (CCG). The method also uses projected gradient ascent to solve the formulated subproblem. Experiments on two problems demonstrate the advantages of this method over conventional CCG approaches.

**Strengths:**

The primary contribution of this work is the innovative application of VAE to construct a tighter uncertainty set, thereby reducing over-conservatism in high-dimensional decision-making, which is then addressed through CCG. The paper is clearly presented, with informative visuals such as Figure 2, and well-organized notation and formulations. The experimental results highlight the promise of the proposed method.

**Weaknesses:**

The proposed approach involves training a VAE, whose performance might be sensitive to hyperparameters, computational resources, and the amount of available training data. Additional experiments and discussion could enhance the paper’s applicability. Please see the following questions for further details.

**Questions:**

- In Figure 1, the authors use two 3D visualizations to illustrate the two-stage operations on $\mathcal{U}$ and $\mathcal{Z}$. Could the authors provide a brief description of the specific main problem addressed here to give the audience a better understanding?

- In the experiment on the production-distribution problem, the authors observed a reverse effect of the bottleneck dimension in the low-dimensional case of $|\mathcal{J}|=3$. Could the authors elaborate on possible reasons for this?

- Based on the experiments, could the authors provide practical guidelines for selecting the appropriate bottleneck dimension for VAEs according to the dimensionality or complexity of the uncertainty set?

- For tabular results such as those in Figure 3 (left) and Table 1, could the authors also report the standard deviation across trials? This would help readers understand the robustness of AGRO in different practical scenarios.

- Could the authors provide more details on the VAE architecture and training settings used in each experiment? Such as layer dimensions, normalization, optimizer, and learning rate,  for better reproducibility.

---

> ### Author Response · Authors · 2024-11-23
> **Clarifying Figure 1**
>
> We appreciate the reviewer's suggestion to refine the description of Figure 1 to convey the nature of the main problem better. We revised the description to include a reference to Equation 3 in Section 2.1 and changed the sentence "First-stage decisions $\boldsymbol{x}^*$ are obtained by solving a main problem for a finite set of uncertainty realizations, $\mathcal{S}$" to "First-stage decisions $\boldsymbol{x}^*$ are obtained by solving a main problem, which approximates the original ARO uncertainty set $\mathcal{U}$ by a finite scenario set $\mathcal{S}$ (see Eq. (3))".

---

> ### Author Response · Authors · 2024-11-23
> **Role of bottleneck dimension**
>
> Regarding the reviewer's comment on providing practical guidelines for choosing the bottleneck dimension, we first kindly refer the reviewer to our response to Reviewer vrQ2, entitled ``Role of bottleneck dimension.'' Building on this, we recommend selecting the bottleneck dimensionality by considering: (1) quantitative generative model metrics (e.g., density and coverage), (2) qualitative evaluations of generated samples (e.g., visual inspection), and (3) cost estimates associated with the downstream ARO objective.
>
> Specifically, when a large amount of data is available to obtain robust out-of-sample estimates of planning costs, we suggest selecting $L$ to be as small as possible while still achieving a reliable approximation of the chance constraint. This can be done by comparing $q(\boldsymbol{\xi}, \boldsymbol{x}^*)$ for $\boldsymbol{\xi}$ obtained from the adversarial subproblem against sample estimates $\hat{F}^{-1}(\alpha; \boldsymbol{x}^*)$ (see Fig.~5 in Appendix B.2.1). Conversely, in settings where out-of-sample evaluations of planning costs are difficult to obtain (e.g., if data is limited or solving the recourse problem is costly), we recommend selecting $L$ to achieve satisfactory diversity and fidelity metrics. For example, $L$ should be chosen as small as possible while ensuring that density scores remain above a minimum threshold, such as $0.9$, to achieve a safer approximation of the chance constraint.
>
> The reviewer also raises an insightful point regarding the observed reversed effect of bottleneck dimensionality for $|\mathcal{J}| = 3$. We attribute this to the same phenomenon that led to lower planning costs for $L = 2$ in the capacity expansion study. Specifically, reduced coverage of the uncertainty set (evident for $L = 1$ when $|\mathcal{J}| = 3$) offsets the inherently conservative nature of the uncertainty set's approximation of chance constraints, ultimately resulting in lower planning costs. However, as $|\mathcal{J}|$ increases, models with $L = 4$ outperform $L = 1$ as low fidelity and insufficient coverage lead to increased costs due to underconservatism. We elaborate on this phenomenon in our comment entitled ``Role of bottleneck dimension'' in response to Reviewer vrQ2.
>
> In light of several reviewers' comments regarding the role of the VAE bottleneck dimension -- both generally and with regard to the experimental results of the production distribution problem -- we are currently revising Section 4 of the text to more clearly convey these findings in relation to observations from both experimental studies.

---

> ### Author Response · Authors · 2024-11-23
> **Additional Experimental Details**
>
> We appreciate the reviewer's suggestion to include additional experimental details to enhance the completeness and reproducibility of our work. For Figure 3 (left), we will provide the standard deviations from the 50 previously conducted experimental trials in the revised manuscript. Regarding Table 1, we plan to perform additional experiments for the capacity expansion study to obtain more robust estimates of average costs, runtimes, and their respective standard deviations. Furthermore, we will expand the Computational Details subsections in Appendix B.1 and B.2 to include more comprehensive information about the VAE architecture and training setup.

---

### Official Review · Reviewer_vrQ2 · 2024-11-12

**Soundness:** 3
**Presentation:** 3
**Contribution:** 2
**Rating:** 5
**Confidence:** 3

**Summary:**

This paper introduces AGRO, a novel method for two-stage adaptive robust optimization (ARO) using a variational autoencoder (VAE) to generate adversarial and realistic uncertainty sets. The authors demonstrate that AGRO reduces planning costs in ARO tasks, outperforming classical approaches.

**Strengths:**

1. The proposed AGRO framework is innovative, embedding a VAE within a column-and-constraint generation (CCG) scheme to achieve high-dimensional adversarial generation with cost efficiency.
2. It seems that the empirical results highlight an over 10% cost reductions over classical methods.

**Weaknesses:**

1.	The introduction lacks a comprehensive motivation for using a VAE for uncertainty sets over other generative models. The authors should justify why a VAE was chosen and discuss the potential advantages over alternatives, like GANs or normalizing flows, which may also be suitable.
2.	The discussion on the choice of the VAE bottleneck dimension (parameter L) could be expanded. The authors should provide more insight into how different L values affect the uncertainty set’s coverage and the balance between computational cost and model fidelity.
3.	While the experiments are detailed, there is no mention of computational time for VAE training or comparison with other ARO solutions. Including such results would enhance transparency about AGRO’s feasibility in larger-scale applications.
4.	The paper does not explore alternative formulations for the adversarial subproblem. A comparison with different optimization methods or a discussion on the limitations of projected gradient ascent could further clarify AGRO's robustness.

**Questions:**

1.	Why did you choose a VAE over other generative models (e.g., GANs, normalizing flows) for constructing uncertainty sets in AGRO? Would these models offer any advantages or limitations compared to VAEs in this application?
2.	How does the bottleneck dimension (L) influence the overall performance and reliability of AGRO? Could you elaborate on any trade-offs between computational cost and uncertainty set coverage as L varies?
3.	The paper discusses using VAE-based uncertainty sets to achieve tighter approximations. Could you clarify how you ensure these sets are both realistic and adversarial? Are there any specific quantitative or qualitative metrics that assess the accuracy of these generated uncertainty sets?

---

> ### Author Response · Authors · 2024-11-23
> **Motivation for using VAEs**
>
> We chose VAEs over other generative methods such as GANs and normalizing flows due to their relatively high training stability and low computational cost for sampling. These characteristics make VAEs particularly well-suited for integration within the AGRO framework. This modeling choice is detailed in Section 3.1. While GANs and normalizing flows can offer potential advantages, particularly in generating higher-fidelity samples in high-dimensional settings, we found that these methods presented notable challenges. Specifically, during preliminary experiments on the capacity expansion case study, neither GANs nor normalizing flows produced samples with significantly better fidelity than the VAE. Moreover, these methods were more challenging to train and, in the case of normalizing flows, slower with regard to generating samples. Consequently, we decided against conducting a comprehensive comparison between these approaches.
>
> However, we acknowledge that VAEs may encounter limitations in generating high-fidelity samples in more complex, high-dimensional settings. To reflect this, the revised version of Section 3.1 offers a more balanced discussion of the trade-offs and limitations of VAEs compared to other generative methods. Specifically, we add that very high-dimensional settings may necessitate the use of alternative generative modeling approaches, such as GANs or normalizing flows, that are known to achieve higher fidelity than VAEs.

---

> ### Author Response · Authors · 2024-11-23
> **Role of bottleneck dimension**
>
> We thank the reviewer for their insightful observation. Below, we elaborate on our findings, which we are working to incorporate as revisions to the main text.
>
> In our experiments, we observed that the optimal choice of bottleneck dimension depended on the problem setting. Interestingly, lower-dimensional bottlenecks sometimes resulted in reduced costs despite yielding worse diversity and fidelity metrics. We hypothesize that this behavior arises due to two factors:
>
> 1. Uncertainty set-based approximations of chance-constrained programs are inherently conservative. These sets aim to capture a region containing 95\% probability mass, rather than the region containing the least adversarial realizations, which would provide the tightest approximation of the chance constraint.
>
> 2. Reduced fidelity can lead to less adversarial realizations, thereby lowering costs driven by over-conservatism. For example, a VAE with a low-dimensional bottleneck may fail to achieve the desired 95\% coverage, but the reduced coverage offsets the conservative nature of the uncertainty set approximation, ultimately leading to lower planning costs.
>
> This phenomenon was evident in our capacity expansion planning experiments, where the "smoothing" effect of a VAE with $L=2$ resulted in lower planning costs compared to higher-dimensional bottlenecks ($L > 2$), which produced more volatile load profiles. A similar trend was observed in the production distribution problem. For a low-dimensional setting ($|\mathcal{J}|=3$), the VAE with $L=1$ achieved lower costs compared to $L=4$. However, in a higher-dimensional setting ($|\mathcal{J}|=12$), the VAE with $L=4$ outperformed $L=1$. In this regime, low fidelity led to higher costs due to underconservatism, in contrast to the lower-dimensional case.
>
> These findings indicate that the bottleneck dimension $L$ should not be optimized solely for diversity and fidelity metrics. Instead, $L$ can be tuned to balance these metrics with the performance of first-stage decisions on a held-out set. In data-scarce settings, prioritizing satisfactory diversity and fidelity metrics may be preferable, emphasizing uncertainty set coverage over potentially nonrobust planning cost estimates. Ultimately, this tradeoff requires users to consider a combination of quantitative generative model metrics, qualitative sample evaluations, and performance on the downstream ARO objective.
>
> In response to reviewer comments regarding the role of the VAE bottleneck dimensionality, the final manuscript will include revisions to Section 4 that (1) consolidate observations from both case studies to make the above conclusions regarding the impact of $L$ more explicit and (2) recommend best practices for choosing $L$.

---

> ### Author Response · Authors · 2024-11-23
> **Ensuring accuracy of generated uncertainty sets**
>
> We appreciate the reviewer’s concern regarding the ability of our approach to ensure that the generated uncertainty sets are appropriately adversarial and realistic. While we acknowledge that our findings are not necessarily generalizable to all problem settings, we emphasize that theoretical guarantees are outside the scope of this work. Instead, our focus lies on experimentally demonstrating the efficacy of the proposed approach for reducing planning costs in ARO.
>
> To evaluate the realism and coverage of the generated uncertainty sets, we employ both qualitative and quantitative assessments of the generative model's performance. Quantitatively, we rely on standard metrics such as precision, density, recall, and coverage scores [1] to evaluate the diversity and fidelity of the VAE. Qualitatively, we compare visualized samples generated by the model to true observations, providing additional evidence of the realism of the scenarios produced.
>
> While these methods do not offer explicit guarantees on the coverage of the generated uncertainty sets, our experimental results strongly suggest that AGRO generates appropriately adversarial realizations. Specifically, our capacity expansion study demonstrates that AGRO yields robust approximations of the 95\% chance constraint in all tested cases except $L=2$. This conclusion is substantiated by Fig. 5 in Appendix B.2.1, where the worst-case costs estimated by AGRO consistently exceed the sample-based estimates -- used as proxies for the true objective -- except for $L=2$. Furthermore, Fig. 4 shows that worst-case realizations derived from the AGRO uncertainty set are more realistic compared to those obtained using classical uncertainty sets.
>
> We recognize the importance of providing a balanced and transparent discussion of these findings. To address this, Section 4 of
> the final manuscript will include revisions that make the above conclusions more explicit alongside discussion of best practices for choosing $L$ (see our response entitled "Role of bottleneck dimension").
>
> References:
>
> [1] Naeem, Muhammad Ferjad, et al. "Reliable fidelity and diversity metrics for generative models." International Conference on Machine Learning. PMLR, 2020.

---

> ### Author Response · Authors · 2024-11-23
> **Comparison of PGA with other methods**
>
> Indeed, a comparison with alternative algorithms is useful for benchmarking the robustness and performance of AGRO. In this context, we attempted to solve the bilinear formulation of the adversarial subproblem (see Equation 7 in Appendix A.2) for the capacity expansion problem using Gurobi's nonconvex optimizer. Additionally, we explored precomputing upper bounds for the variables $\boldsymbol{z}^{(\ell)}$ and $\tilde{\boldsymbol{z}}^{(\ell)}$ as suggested in [1] to enhance scalability. Despite these efforts, we were unable to obtain certificates of optimality for the subproblem within a reasonable time. For example, after one hour of computation, the solver reported an optimality gap of 400\%. Given the high computational cost of solving this subproblem in each iteration of CCG, we decided to discontinue this approach.
>
> In response to feedback from reviewers requesting comparisons between PGA and other solution methods, we revisited this approach, aiming to reduce runtimes by providing a warm start for the bilinear formulation using the solution obtained via PGA. Unfortunately, even with this enhancement, we could not achieve a provably optimal solution for a single iteration of the adversarial subproblem within the one-hour time limit.
>
> For completeness, we are generating additional results from applying Gurobi's nonconvex optimizer to solve the adversarial subproblem for the production distribution problem, which is considerably lower dimensional (and perhaps more tractable) than the capacity expansion problem. The final submission will include findings regarding the optimality of solutions obtained with PGA compared to provably optimal solutions, should they be achievable. In the case that provably optimal solutions are not achievable (as was the case for the capacity expansion study), the final submission will include a brief summary of these findings.
>
> References:
>
> [1] Fischetti, Matteo, and Jason Jo. "Deep neural networks and mixed integer linear optimization." Constraints 23.3 (2018): 296-309.

---

> ### Author Response · Authors · 2024-11-23
> **VAE training time**
>
> Regarding training times for the VAEs, we provide results for the production distribution study in the Computational Details subsection of Appendix B.1 and for the capacity expansion study in Tab. 1 in Section 4.2.

---

### Author Response · Authors · 2024-12-02
**Final note to reviewers**

As we approach the conclusion of the discussion period, we invite reviewers to share any final questions or comments for the authors before proceeding with their evaluations. Below is an overview of new findings and major revisions made to the final submitted manuscript:

### Revised Deterministic Parameters in Section 4
We have updated the deterministic parameters used to define the capacity expansion problem in Section 4, enhancing the realism of the capacity expansion model. This adjustment is reflected in the latest experimental findings, where the case of $L=4$ now yields the lowest cost across all methods instead of $L=2$. This shift highlights the influence of ramping and net-load volatility on operational and overall planning costs. The revised parameters also resulted in a less tractable bilinear subproblem for CCG, leading to a sixfold increase in runtime. These refinements provide a more faithful representation of capacity expansion planning considerations, ultimately strengthening our experimental results.

### New Subsection in Section 4.2
We have added a subsection titled "Bottleneck Dimension" to Section 4.2. This subsection connects insights on the role of the bottleneck dimension $L$ from the production-distribution case study to those from the capacity expansion planning case study. Additionally, it includes a paragraph offering recommendations for selecting $L$.

### Updates to Section 3.1
The second paragraph of Section 3.1 has been revised to highlight the potential advantages of GANs, normalizing flows, and diffusion models in generating higher-fidelity samples compared to VAEs. We then justify the use of VAEs by referencing experimental findings in Section 4, which show that higher generative fidelity does not necessarily lead to better performance concerning the ARO objective.

### Expanded Computational Experiment Details
Additional details have been included regarding the computational experiments. Specifically, Figure 3 (left) and Table 1 now report standard deviations across all trials. Furthermore, Appendices B.1 and B.2 have been expanded to provide more comprehensive information about the architecture and training process for the VAEs.

### Comparison of PGA with Direct Solve
Finally, we note that the final submission does not include a complete comparison of projected gradient ascent (PGA) with the direct solve approach for the mixed-binary bilinear formulation of the AGRO subproblem (see Appendix A.2). However, our experimental findings can be summarized as follows:

- For the production-distribution problem with $|\mathcal{J}| \in \{3,6\}$, PGA obtained solutions to the adversarial subproblem that were, on average, within 1\% of optimality as determined by the direct solve.

- For the capacity expansion planning problem, a full cost comparison could not be provided as the direct solve failed to achieve provably optimal solutions within a reasonable timeframe. Specifically, the solver reported a 400\% optimality gap after one hour.

***

### Thank you!
We are grateful for your thoughtful feedback throughout this process. We look forward to your final evaluations and appreciate your valuable contributions to improving this manuscript.

---

### Meta-Review · Area_Chair_ziyt · 2024-12-29

**Metareview:**

This paper introduces AGRO, a two-stage adaptive robust optimization (ARO) method that combines variational autoencoder (VAE) within a column-and-constraint generation for adversarial uncertainty sets. The authors show that AGRO reduces planning costs compared to classical ARO approaches.

This is indeed a borderline paper. Although the proposed method achieves significant cost reduction, the exploiting generative model in learning for decision making and RL is not novel, and the choice of VAE seems not well justified.

**Additional Comments On Reviewer Discussion:**

The authors clarified most questions raised by the reviewers. Most of the reviewers acknowledge the contribution of the paper and achieved an agreement, recognizing the paper as borderline.

---

### Decision · Program_Chairs · 2025-01-22

Accept (Poster)